# Saving less in China facilitates global CO$_2$ mitigation

Chen Lin[1,6], Jianchuan Qi[2,6], Sai Liang [2✉], Cuiyang Feng[2], Thomas O. Wiedmann [3], Yihan Liao[1], Xuechun Yang[2], Yumeng Li[2], Zhifu Mi [4] & Zhifeng Yang[2,5]

Transforming China's economic growth pattern from investment-driven to consumption-driven can significantly change global CO$_2$ emissions. This study is the first to analyse the impacts of changes in China's saving rates on global CO$_2$ emissions both theoretically and empirically. Here, we show that the increase in the saving rates of Chinese regions has led to increments of global industrial CO$_2$ emissions by 189 million tonnes (Mt) during 2007–2012. A 15-percentage-point decrease in the saving rate of China can lower global CO$_2$ emissions by 186 Mt, or 0.7% of global industrial CO$_2$ emissions. Greener consumption in China can lead to a further 14% reduction in global industrial CO$_2$ emissions. In particular, decreasing the saving rate of Shandong has the most massive potential for global CO$_2$ reductions, while that of Inner Mongolia has adverse effects. Removing economic frictions to allow the production system to fit China's increased consumption can facilitate global CO$_2$ mitigation.

[1] School of Applied Economics, Renmin University of China, Beijing 100872, P. R. China. [2] State Key Joint Laboratory of Environment Simulation and Pollution Control, School of Environment, Beijing Normal University, Beijing 100875, P. R. China. [3] School of Civil and Environmental Engineering, The University of New South Wales, Sydney, NSW 2052, Australia. [4] The Bartlett School of Construction and Project Management, University College London, London WC1E 7HB, UK. [5] Institute of Environmental and Ecological Engineering, Guangdong University of Technology, Guangzhou, Guangdong 510006, P. R. China. [6]These authors contributed equally: Chen Lin, Jianchuan Qi. ✉email: liangsai@bnu.edu.cn

In 2017, China's domestic saving rate was ~47% of its Gross Domestic Product (GDP), one of the highest in the world[1]. Such a high saving rate accelerates investment and prompts capital accumulation for economic growth. It also affects $CO_2$ emissions in China and the world as a whole. From the early 1980s to the early 2000s, China has witnessed a rapid increase in its national saving rate[2,3]. The changes in saving and investment rates in China are synchronous in most years[4]. Thus, a high saving rate in China implies a high investment rate. The saved capital stimulates the demand for investments, including in infrastructure, equipment and machinery, which are mostly energy-intensive products from the life cycle perspective[5–7]. Consequently, a higher saving rate leads to higher $CO_2$ emissions in China.

The central aim of the Paris Agreement is to keep the increase in global temperature well below 2 °C above pre-industrial levels in this century and to pursue efforts to limit the temperature increase even further to 1.5 °C[8]. Among the 170 nations signing the Paris Agreement, China has declared a $CO_2$ emission peak target in 2030. Meanwhile, although the high investment prompted economic growth in China, persistent high investment suffers from the decreasing rate of return to capital and would harm the growth of household welfare in the long run. Therefore, a growth mode relying on high investment is unsustainable. Against this background, China has recently launched initiatives to transform the economic growth pattern from investment-driven to consumption-driven. The most recent policy is "Promoting the upgrade of key consumer goods and smoothing the implementation of resource recycling (2019–2020)" by the National Development and Reform Commission, Ministry of Ecology and Environment and Ministry of Commerce of China on June 6th, 2019. The consumption promotion initiatives will change the saving rates in China. Meanwhile, the ageing process in China is also expected to reduce the saving rates in China[9]. Because of the massive amounts of capital involved, it is urgent to observe the extent to which the changes in saving rates of China can help both the world and China achieve $CO_2$ mitigation targets. Unfortunately, existing studies have not investigated this issue. The insights from empirical work in this study will help us understand a new potential source of $CO_2$ mitigation in the world.

The role of China in global $CO_2$ emissions has been widely considered in existing studies. Many studies focus on the impacts of structural changes in the final demand of China on global or regional $CO_2$ emissions[10–14]. Specifically, scholars have taken into account the influence of China's final consumption structure or investment structure on $CO_2$ emissions[15–18]. However, to the best of our knowledge, the impacts of changes in China's saving rates on global $CO_2$ emissions have not been investigated. Meanwhile, saving rates in China vary significantly across regions, from 37% in Shanghai to 70% in Inner Mongolia in 2012 (calculated in this study). This variance implies that the $CO_2$ mitigation potential of consumption promotion policies in China will vary across regions. Thus, measuring these different potentials by considering regional diversity is also crucial for region-specific policy decisions.

In this study, for the first time, we propose a new structural decomposition analysis (SDA) method with the global multi-regional input–output (MRIO) model to analyse the partial effects of the changes in the saving rates of Chinese regions on $CO_2$ emissions from global production systems (named as global industrial $CO_2$ emissions) from the perspective of comparative statics. Further, based on the SDA, we conduct a counterfactual analysis to show the effects of the potential decrease in saving rates of China on global $CO_2$ mitigation. Results show that, from 2007 to 2012, the increase in the saving rates of Chinese regions

has led to global industrial $CO_2$ emission increments by 189 million tonnes (Mt), if other factors remain constant. As a result of the counterfactual analysis, ceteris paribus, a 15-percentage-point decrease in the saving rate of Chinese regions can lower global $CO_2$ emissions by 186 Mt, or 0.7% of global industrial $CO_2$ emissions. Greener consumption in China can lead to a further 14% reduction in global industrial $CO_2$ emissions. This study links growth pattern transformation with global $CO_2$ mitigation. It provides a new tool for achieving the goals set in the Paris Agreement and China's $CO_2$ emission peak target of 2030. The results featuring regional disparity will help policymakers find region-specific solutions for $CO_2$ mitigation.

## Results
**The partial effects of saving rate changes.** From 2007 to 2012, China's national saving rate has increased from 44 to 53% (calculated in this study). A theoretical dynamic model shows that the saving rate change would reduce $CO_2$ emissions if the cumulative $CO_2$ emission intensity of the production process of capital goods is much larger than that of the consumption goods (details in the Methods section). To empirically prove the conclusion of the theoretical model and quantitatively evaluate the impacts, we calculate the multi-regional and multi-sectoral impacts of saving rates decreases using a newly proposed decomposition model. Results show that the saving rate increase contributed 188.6 Mt to changes in global industrial $CO_2$ emissions during 2007–2012 if other factors remained constant, which accounted for 0.7% of global industrial $CO_2$ emissions in 2012. The effects of saving rate increments on global $CO_2$ emission changes varied across sectors. For nations other than China (Fig. 1a), China's saving rate increments increased $CO_2$ emissions in the sectors of energy generation, resource extraction and manufacturing industries, such as electricity (11.2 Mt), ferrous metals (2.3 Mt) and other minerals (2.1 Mt), if other factors remained constant. The increase of saving rates in China reduced $CO_2$ emission in sectors of agriculture and agricultural products, such as oil seeds (−0.33 Mt), vegetable oils and fats (−0.09 Mt), and other food products (−0.06 Mt), if other factors remained constant. For Chinese regions (Fig. 1b), saving rate increments in China increased $CO_2$ emissions in the sectors of basic material manufacture and construction, such as metallurgy (130.5 Mt), nonmetal products (33.7 Mt) and construction (9.9 Mt), if other factors remained constant. On the contrary, the increase of saving rates in China reduced $CO_2$ emissions in agricultural and service sectors, such as agriculture (−12.8 Mt), food processing and tobaccos (−7.6 Mt) and other services (−7.4 Mt), if other factors remained constant. Since the changes in the saving rates and capital formation rates in China were synchronous in most years (Supplementary Fig. 1), the saving rate increase in China has led to the increments of capital formation and decrements of final consumption in China. It can be observed that the increase of saving rates in China led to $CO_2$ emission increments in sectors mainly providing capital goods, and contributed to $CO_2$ emission reduction in sectors as consumption goods providers.

The effects of saving rate increase in China varied across nations and Chinese regions (Fig. 2). China's saving rate increase led to 168.6 Mt of domestic industrial $CO_2$ emission increments. For other nations, China's saving rate increments increased $CO_2$ emissions of Japan (2.7 Mt), South Africa (1.9 Mt), Australia (1.8 Mt), South Korea (1.7 Mt), India (1.5 Mt) and the United States of America (USA) (1.3 Mt), if other factors remained constant. On the other hand, the saving rate increase in China reduced $CO_2$ emissions of Argentina (−0.2 Mt), New Zealand (−0.02 Mt) and Uruguay (−0.01 Mt), if other factors remained constant. The increase of capital formation in China was a driving force for $CO_2$

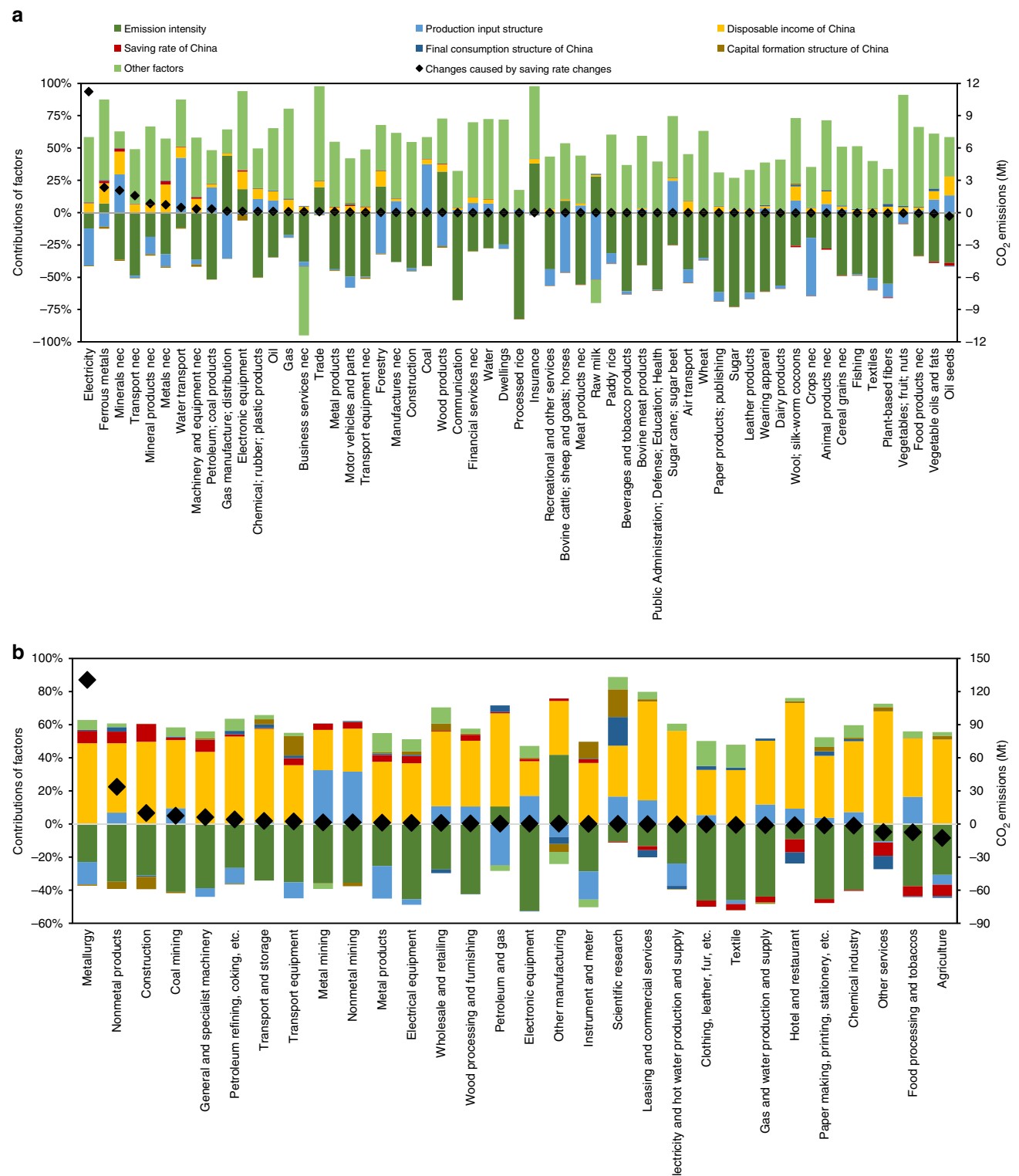

**Fig. 1 Relative contributions of socioeconomic factors to global industrial CO₂ emission changes during 2007–2012. a**, **b** show the relative contributions of socioeconomic factors to global industrial $CO_2$ emission changes in sectors of nations other than China and sectors of China, respectively. The red portion in each bar indicates the relative contribution of saving rate changes in China during 2007–2012. The mark (black diamond) indicates the amount of $CO_2$ emission changes caused by saving rate increments in China during 2007–2012.

emission increments in nations providing intermediate production materials to China. Japan, South Korea and the USA were critical suppliers of machinery and electronic equipment to China, while South Africa, Australia and the Russian Federation (Russia) provided minerals and fossil fuels to China.

Meanwhile, the decrease in China's final consumption was a driving force reducing $CO_2$ emissions in the exporters of agricultural products (e.g., Argentina and New Zealand) to China.

Figure 2 also shows regional disparity within China. China's saving rate increments have increased $CO_2$ emissions in most

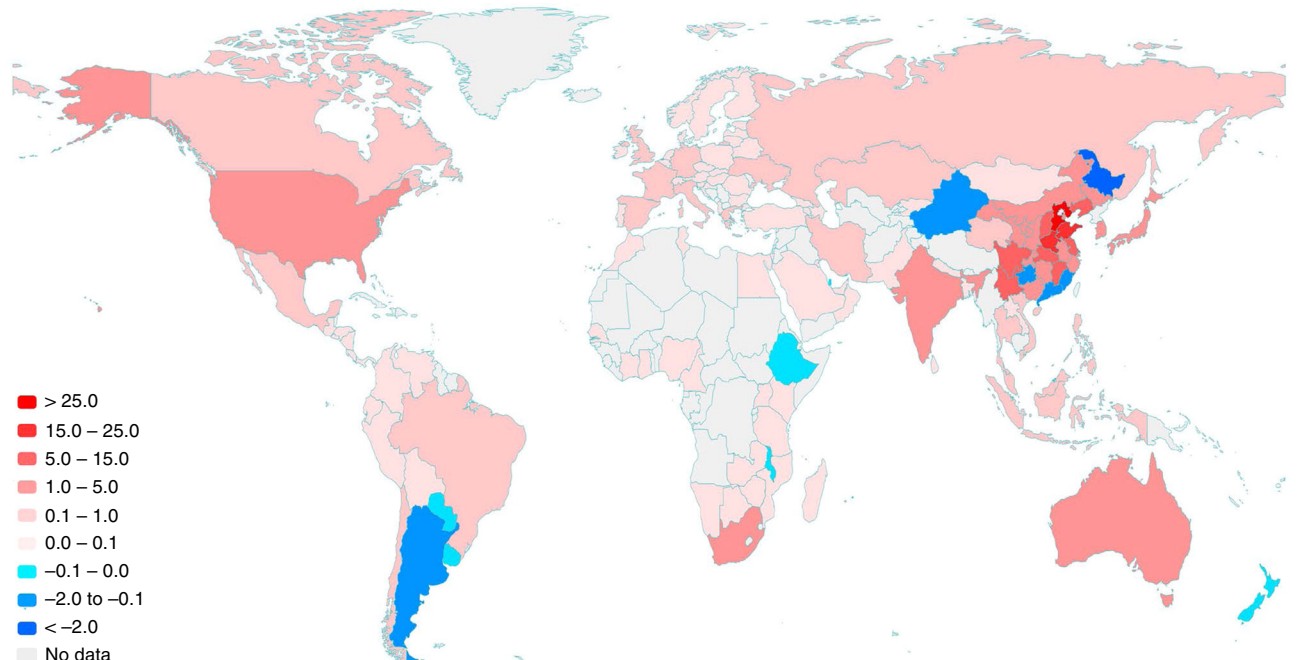

**Fig. 2 Changes in CO$_2$ emissions of nations and Chinese regions caused by saving rate changes in China during 2007–2012 (Mt).** CO$_2$ emission changes caused by saving rate increase in China varied across nations and Chinese regions. Saving rate changes in China caused CO$_2$ emission increase in regions of red colour, while caused CO$_2$ emission decrease in regions of blue colour. A darker colour indicates a larger CO$_2$ emission change.

Chinese regions, such as Hebei (26.3 Mt), Shandong (23.4 Mt), Henan (19.5 Mt), Liaoning (14.7 Mt) and Jiangsu (14.7 Mt), if other factors remained constant. Meanwhile, the saving rate increase in China has reduced CO$_2$ emissions in only five provinces, which were Heilongjiang (−5.5 Mt), Xinjiang (−1.9 Mt), Fujian (−0.7 Mt), Guizhou (−0.3 Mt) and Guangdong (−0.3 Mt), if other factors remained constant. The increase of capital formation in China led to CO$_2$ emission increase in regions with heavy industries (e.g., Hebei and Liaoning). In contrast, the decline of final consumption led to CO$_2$ emission decrease in food-producing regions (e.g., Heilongjiang) and textile-producing regions (e.g., Guangdong).

The SDA with historical data proves the positive correlation between China's saving rates and global CO$_2$ emissions. This finding has substantial policy implications because China's saving rates began to decrease since 2012 and will continue to decrease due to the current consumption promotion policy. However, we could not use SDA with historical data to illustrate the situation with the decreasing trend of saving rates in China and uncover the impacts of different regions in China, because the latest version of Chinese MRIO data is for the year of 2012. In the years when Chinese MRIO tables are published, we can only witness an increasing process of saving rates in China. However, we have proved the positive correlation between China's saving rates and global CO$_2$ emissions with historical data. Hence, we do not need to consider the non-Leontief properties (e.g., factor substitution) of a general equilibrium model for the counterfactual analysis, because they have been considered in the SDA. Therefore, we use the Leontief quantity model that corresponds with the SDA model to calculate the partial effects of the decrease in saving rates. To see the direction and magnitude of the changes in global CO$_2$ emissions, we evaluated the effects of the decrease in China's saving rate by conducting a counterfactual analysis. To keep the analysis univariate and clear, the other factors (e.g., global supply chains and CO$_2$ emission intensity) are kept constant (see Methods section).

**Counterfactual analyses for future evaluation purposes.** We consider four counterfactual scenarios (details in the Methods section). In Scenario 1, a 15-percentage-point decrease in the saving rate of Chinese regions can reduce 323 Mt (1.2%) of global industrial CO$_2$ emissions, but increase CO$_2$ emissions of Chinese households by 137 Mt because of the increased final consumption (more results in Supplementary Data 1). Figure 3 shows the changes in global CO$_2$ emission amounts for nations and Chinese regions, while Supplementary Fig. 2 shows the changes in percentage ratios. The quantity results highlight eastern regions in China and nations with large economic scales. In contrast, percentage-ratio results highlight western regions in China and nations with relatively small economic scales (e.g., nations in South America, Southeast Asia and Africa).

China's CO$_2$ emissions decrease by 153 Mt, which is 1.7% of China's total CO$_2$ emissions in 2012. CO$_2$ emissions from production systems in Japan, South Africa and South Korea decrease by 4.1 Mt (0.4%), 3.9 Mt (1.2%) and 2.9 Mt (0.6%), respectively, and those in the USA decrease by 1.6 Mt (0.04%). Moreover, CO$_2$ emissions in Russia and Germany decrease by 1.5 Mt (0.1%), and 1.3 Mt (0.2%), respectively, making them the top two European nations with the largest CO$_2$ emission reductions. Argentina, New Zealand and Qatar are the top three nations, with increases in CO$_2$ emissions of 0.27, 0.14 and 0.05 Mt, respectively. From the perspective of percentage changes in CO$_2$ emissions from production systems, Zambia (−2.9%), Chile (−1.4%) and South Africa (−1.2%) are the top three nations other than China with the largest CO$_2$ emission reductions. In contrast, Uruguay (0.6%), New Zealand (0.5%) and Argentina (0.2%) are the top three nations with the largest CO$_2$ emission increases. Significant decreases in CO$_2$ emissions are observed in nations providing intermediate production materials to China, while CO$_2$ emissions from the exporters of agricultural products to China increase due to the increase in China's final consumption.

In China, we also observe regional disparity. The CO$_2$ emissions of Hebei, Jiangsu, Liaoning and Shandong decrease by 34.2, 27.0, 24.1 and 15.4 Mt, respectively. Ningxia (9.5%),

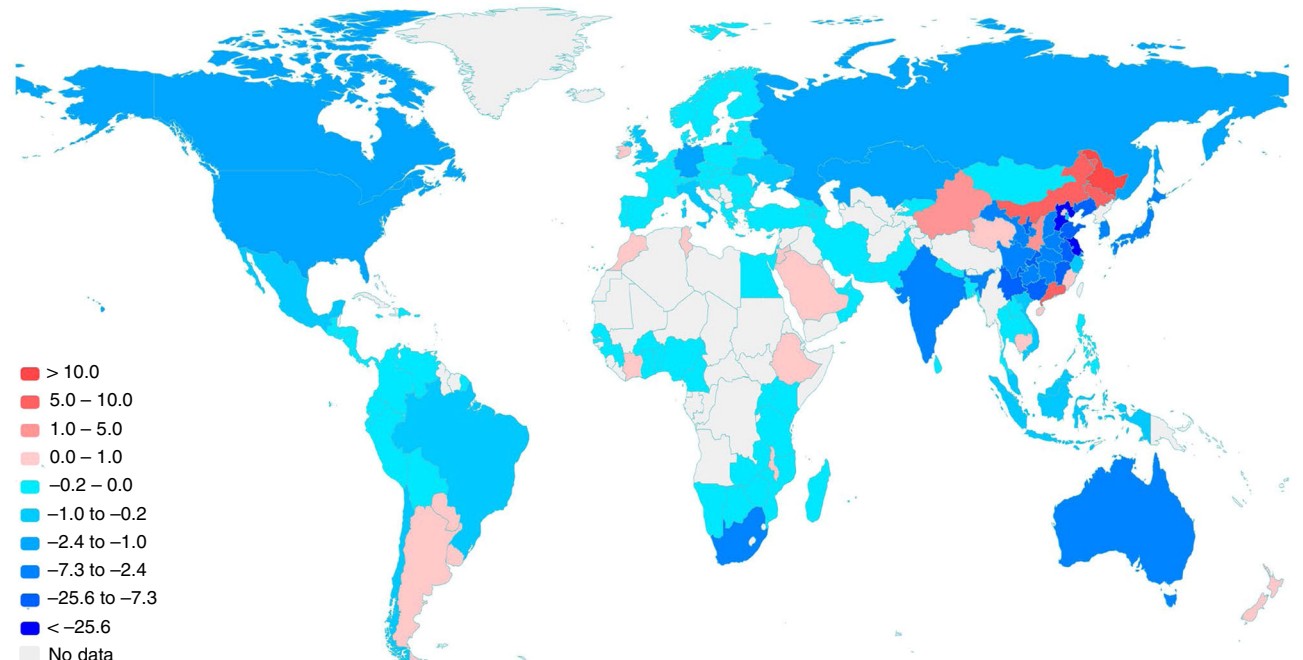

**Fig. 3 Changes in CO₂ emission amounts of nations and Chinese regions (Mt), with conditions of saving rates of all Chinese regions decreasing by 15-percentage points simultaneously and emission intensity as of 2012 (Scenario 1).** Saving rate decrease in China would reduce CO₂ emissions in regions of blue colour, while increase CO₂ emissions in regions of red colour. A darker colour indicates a larger CO₂ emission change.

Chongqing (7.1%), and Jiangxi (7.0%) have the largest reduction rates. Meanwhile, for Heilongjiang, Inner Mongolia and Guangdong, CO₂ emissions increase by 21.4 (8.9%), 9.6 (1.7%) and 7.2 Mt (1.6%), respectively. The CO₂ emission reductions of regions biased towards heavy industries (e.g., Liaoning, Hebei and Gansu) benefit more from the decrease in capital formation than do other regions. Moreover, the CO₂ emissions of regions with abundant mineral resources (e.g., Yunnan and Sichuan) decrease due to reduced capital formation. In contrast, the CO₂ emissions of food-producing regions (e.g., Heilongjiang) and textile-producing regions (e.g., Guangdong) increase due to China's large final consumption of foods and clothes.

To understand the mechanism behind the decrease in global CO₂ emissions due to saving rate reductions, this study calculates the global CO₂ emissions induced by a unit basket of final consumption or capital formation in China (Fig. 4). The global CO₂ emissions induced by a unit of final consumption in China are 1.0 t (more results in Supplementary Data 2 and 3), while those induced by a unit of capital formation in China are 1.5 t (more results in Supplementary Data 4 and 5). The former is lower than the latter, which explains the reductions in global CO₂ emissions from decreases in China's saving rates. China's final consumption of agricultural products, foods, petrochemicals and nonmetal products, machinery, equipment and services mainly drives global CO₂ emissions, whereas China's capital formations primarily lead to global CO₂ emissions through machinery, equipment, construction and services (Fig. 4).

Final consumption in China leads to significant CO₂ emissions in developed nations (notably the USA, South Korea, Japan and Germany) and South and Southeast Asian nations (notably India, Malaysia, Thailand, Indonesia and Viet Nam). China's final consumption of machinery and equipment causes CO₂ emissions in developed nations. South and Southeast Asia discharge CO₂ due to China's final consumption of foods, chemicals and clothes. The capital formation in China causes CO₂ emissions in developed nations in East Asia, Western Europe and North America, as well as developing nations in South and Southeast

Asia. Notably, the USA-China trade in consumer products has larger contributions to global CO₂ emissions than does the USA-China trade in capital goods.

This study also reveals that the sources of goods and services that characterise China's final consumption are more widespread than the sources of China's capital formation of goods and services (Fig. 4). This finding indicates that promoting China's low-carbon consumption can facilitate widespread effects of CO₂ mitigation around the world, while maximizing the CO₂ reduction potential of China's low-carbon capital formation should focus on products of particular limited nation/region-sectors (e.g., machinery, equipment and vehicles from South Korea and Japan; machinery, vehicles and transport equipment from Germany and the USA; and construction, machinery and equipment from Jiangsu and Shandong in China).

The structures of final consumption and capital formation vary across Chinese regions, resulting in different effects on global CO₂ reductions. This study estimates changes in global CO₂ emissions caused by changes in the saving rate (a decrease of 15-percentage points) of individual Chinese regions to uncover the disparity among Chinese regions (Scenario 2, Fig. 5a, more results in Supplementary Data 6). For most Chinese regions, decreases in the saving rate will lead to reductions in global CO₂ emissions. The decrease in the saving rate of Shandong (a province in eastern China) has the most significant effects on global CO₂ emission reductions (29 Mt). In contrast, the decrease in the saving rate of Inner Mongolia will cause a 25 Mt increase in global CO₂ emissions.

To reveal the mechanism of global CO₂ emission changes caused by decreases in the saving rates, this study compares the structures of the final consumption and capital formation of Shandong and Inner Mongolia (Fig. 5b, c, more results in Supplementary Data 7 and 8). Major sectors in Shandong's capital formation (e.g., construction and machinery) generally have higher cumulative CO₂ emission intensities than do major sectors in Shandong's final consumption (e.g., other services and food processing). Consequently, a decrease in Shandong's saving

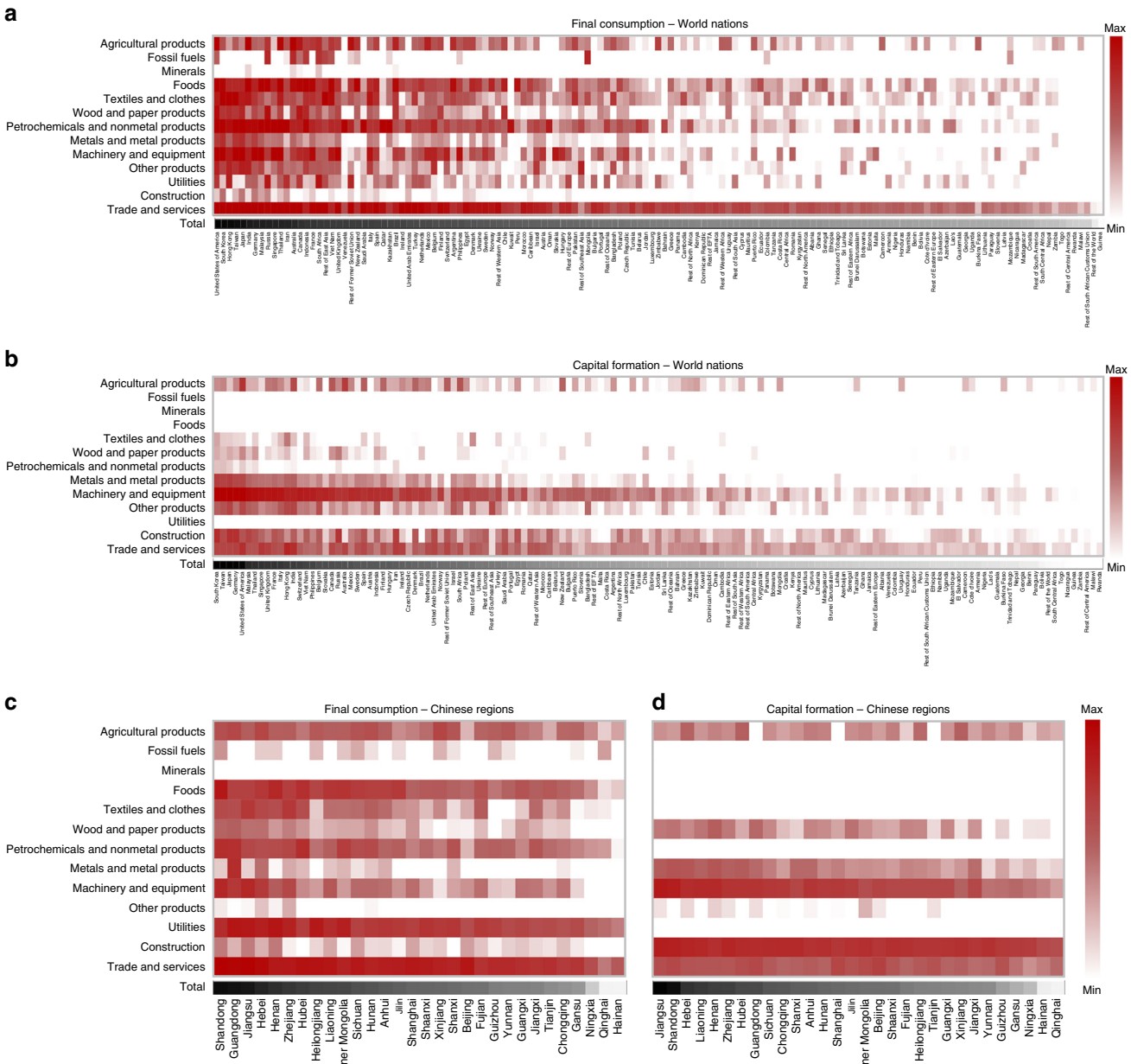

**Fig. 4 Global CO₂ emissions induced by a unit of final consumption or capital formation in China.** Each grid in **a**, **c** indicates the global CO₂ emissions induced by China's one unit of final consumption of products from each sector of nations other than China (**a**) and Chinese regions (**c**). Each grid in **b**, **d** indicates the global CO₂ emissions induced by China's one unit of capital formation of products from each sector of nations other than China (**b**) and Chinese regions (**d**). The black/white bars at the bottom show the total global CO₂ emissions caused by China's one unit of final consumption or capital formation of nations or Chinese regions. A darker colour indicates larger global CO₂ emissions. Results in more sectoral detail are shown in Supplementary Fig. 3.

rate leads to global CO₂ emission reductions. Inner Mongolia has the opposite situation, especially regarding the high cumulative CO₂ emission intensity and the high proportion of electricity in its final consumption. Thus, a decrease in the saving rate of Inner Mongolia will increase global CO₂ emissions.

Shanghai has the lowest saving rate (37%) in 2012, which is close to the forecasted saving rate of China as a whole in 2030[9]. Therefore, Shanghai has the least space to reduce its saving rate. In contrast, Inner Mongolia's saving rate reaches 70%. It clearly has a higher potential for saving rate reduction. To reflect the disparity in the potential of saving rate reductions, this study changes each province's saving rate one by one to 37% (Scenario 3, Supplementary Fig. 4, more results in Supplementary Data 9).

With the saving rates of all Chinese provinces decreasing to 37%, global CO₂ emissions will decrease by 153 Mt (0.6% of global industrial CO₂ emissions), including a 321 Mt decrease from production systems (1.2% of the global total) and a 168 Mt increase from Chinese households. Shandong (35 Mt), Liaoning (18 Mt) and Shanxi (17 Mt) are the top three provinces with the highest potentials for global CO₂ reductions accompanying decreases in their saving rates. In contrast, decreasing the saving rate of Inner Mongolia will increase global CO₂ emissions by 55 Mt.

The previous results from Scenarios 1, 2 and 3 illustrate the CO₂ mitigation potential by reducing China's saving rates. These results also highlight a more prominent role of China's final consumption on global CO₂ reduction. Greener consumption

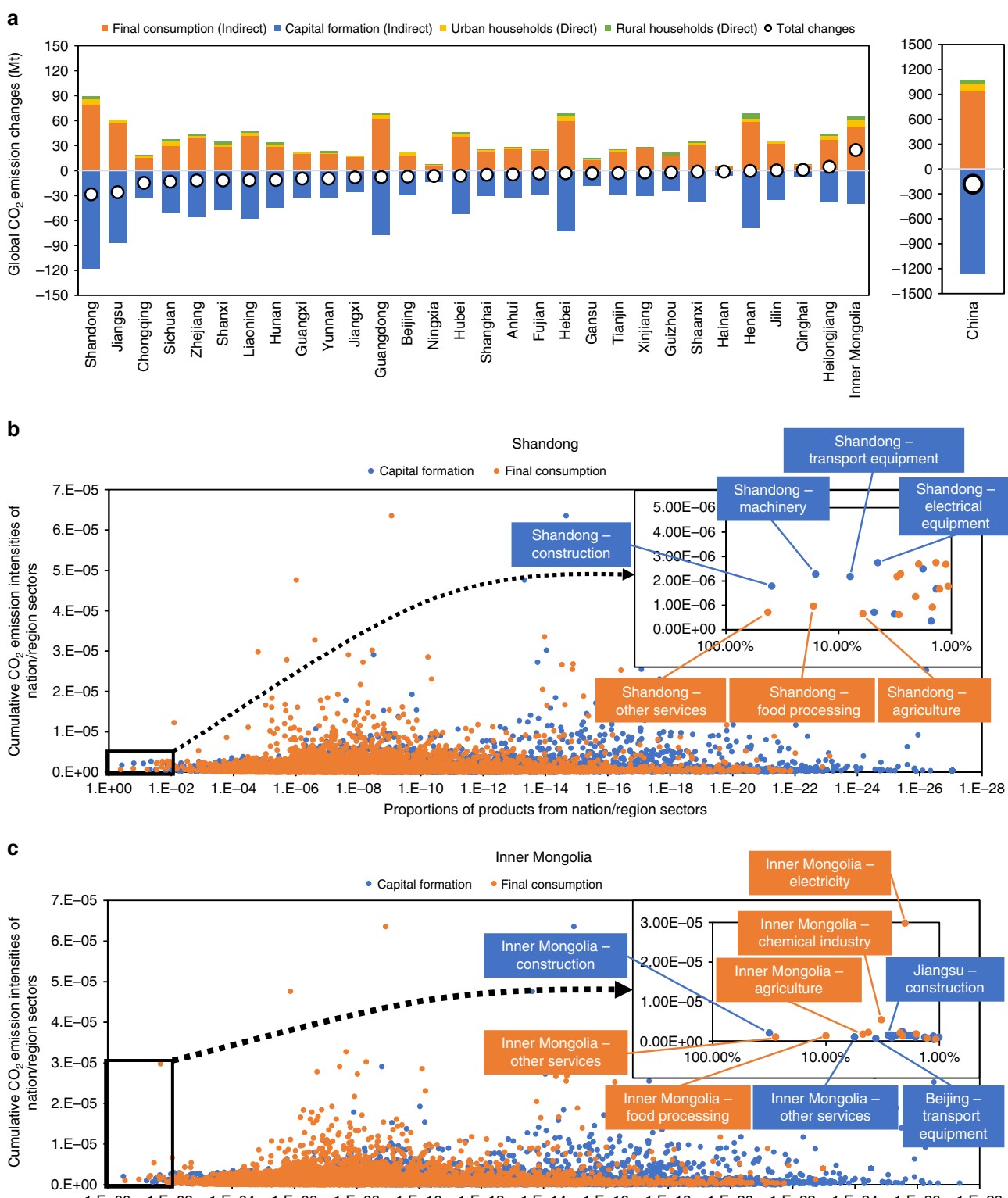

**Fig. 5 Global CO₂ emission changes caused by reducing the saving rate of each Chinese region by 15-percentage points and the particular situations for Shandong and Inner Mongolia.** Panel **a** shows global CO₂ emission changes caused by reducing the saving rate of each Chinese region by 15-percentage points. Panels **b**, **c** show particular situations for Shandong and Inner Mongolia, respectively. The horizontal axis in **b**, **c** represents the proportions of products from nation/region-sectors in final consumption or capital formation, and the vertical axis represents the cumulative CO₂ emission intensities of nation/region-sectors.

**Table 1 Scenario settings in this study.**

| Scenarios | Changes in saving rates | Regional settings | Emission intensity |
|---|---|---|---|
| 1 | Decrease by 15-percentage points | All Chinese regions change simultaneously | As of 2012 |
| 2 | Decrease by 15-percentage points | Only one region changes at a time | As of 2012 |
| 3 | Change to 37% (Shanghai's saving rate in 2012) | Only one region changes at a time | As of 2012 |
| 4 | Decrease by 15-percentage points | All Chinese regions change simultaneously | Best scenario |

initiatives in China will play a more important role in the future. To estimate the potential contribution of China's projects for greener consumption to global $CO_2$ reductions, this study conducts Scenario 4 (Table 1). In this scenario, in addition to the 15-percentage-point reduction in the saving rate, the commodities produced by the most efficient nation with regard to cumulative (both direct and indirect) $CO_2$ emission intensity replace the current commodities consumed in China. We call this replacement "consummate greener consumption". In other words, this scenario shows the lower bound of the global $CO_2$ emissions induced by China's final consumption.

Global $CO_2$ emissions of Scenario 4 are 3593 Mt lower than those of Scenario 1. This finding means that China's "consummate greener consumption" setting leads to 14% of global $CO_2$ reductions, implying the enormous potential of $CO_2$ reductions by promoting greener consumption in China. Within China, reducing the cumulative $CO_2$ emission intensity of the other services (especially in Shandong, Jiangsu, Hebei and Guangdong) and electricity (especially in Henan, Jiangsu, Guangdong and Heilongjiang) sectors has the most significant global $CO_2$ reduction effects (Supplementary Fig. 5). The effects from the food processing sector of Shandong are also large. This result means that increasing the production efficiency of those sectors in China can dramatically reduce global $CO_2$ emissions. For nations that export consumer goods to China, South Korea and the USA show large $CO_2$ reduction potentials (Supplementary Fig. 5).

## Discussion

The results of this study provide new insights into global $CO_2$ mitigation policies and the implementation of the Paris Agreement. We uncover the positive correlation between China's saving rates and global $CO_2$ emissions and prove global $CO_2$ mitigation potentials from reductions in the saving rate in China. Although the main objective of consumption promotion is not environmental protection, it has $CO_2$ reduction co-benefits. Our conclusion is different from certain studies evaluating the $CO_2$ emissions of consumption growth in China[19,20] because our study considers the off-setting effect of final consumption on capital formation.

Although the final consumption in China is cleaner than capital formation, the decrease in the saving rate emphasizes the importance of greener consumption. China should take more initiatives regarding the $CO_2$ mitigation effects of the consumption promotion strategy[21] and the accelerating process of urbanization[20,22]. China should further promote the consumption of low-carbon products in the categories of foods, plastic products, electronic equipment, machinery and electricity, because these products drive massive amounts of $CO_2$ emissions within global supply chains[23,24]. Potential pathways can be establishing a product labelling scheme to indicate the $CO_2$ footprint of consumer goods. Moreover, Chinese governments should remove economic frictions to let the production system fit the increased consumption demand. Countries around the world can help China undertake the economic and social costs of

adjusting the production structure, for example, through international aid and clean development mechanisms.

China is a large country with great regional disparity[13,25]. Thus, region-specific policies are important for China[18]. Existing regional actions on $CO_2$ mitigation are mostly following national plans and have not fully considered regional characteristics of low-carbon consumption potentials. Instead of uniform actions, the Chinese central government should encourage local governments to take region-specific policies matching region-specific low-carbon consumption potentials. For instance, in Shandong, the food industry is crucial for consumption-related $CO_2$ emissions. In Inner Mongolia, however, improving the efficiency of electricity generation is a somewhat high priority. International cooperation (international transfer of technologies) and financial aid from the United Nations to less-developed Chinese regions will help bridge the technology and efficiency gaps among regions.

Uncertainty in the results of this study mainly comes from the uncertainties of the forecasted saving rate changes of China and the global MRIO data. This study estimates the upper and lower ranges of the $CO_2$ emission changes based on the forecasted range of potential saving rate changes in China. The range for global industrial $CO_2$ emission reduction is 292.4–513.3 Mt, while the range for $CO_2$ emission increments of Chinese households is 124.3–218.2 Mt. Thus, the range of total $CO_2$ emissions reduction caused by saving rate changes is 168.1–295.1 Mt (more results in Supplementary Data 10). Improving the accuracy of the forecasting for saving rates in China would reduce the uncertainties of this study. Meanwhile, uncertainties of the global MRIO data would also cause uncertainties in the results. An uncertainty analysis with standard deviations can be used to evaluate the reliability and uncertainty of the results given by a certain database[26]. However, the standard deviations are not given in the Global Trade Analysis Project (GTAP) database used in this study, which makes the condition insufficient for quantitative uncertainty analysis. Besides, unfortunately, due to the lack of uncertainty information on the raw data from the statistical offices, the standard deviations of the MRIO databases are either unavailable or provided by assumptions and choices[27]. If the statistical offices could publish more information on the data variations of their original samples and the degrees of uncertainty in their data processing stages, future work with MRIO models could provide a more precise uncertainty analysis.

We evaluate the sensitivity of the results to all the parameters by calculating their elasticities, which are the ratios between the changing rate of global $CO_2$ emission changes and the changing rate of the parameters. Results show that the elasticities of most parameters are small (Supplementary Fig. 6), which indicates a low sensitivity for the results. The parameter with the highest sensitivity is the $CO_2$ emission intensity of the Metallurgy sector in Hebei (elasticity 0.14), which means that the change in $CO_2$ emissions from global production systems due to the change in saving rates would change by 1.4% if the $CO_2$ emission intensity of the Metallurgy sector in Hebei changes by 10%. Other parameters with relatively high sensitivity include the disposable incomes and saving rate changes of Shandong and Jiangsu. The

$CO_2$ emissions of Chinese households are in a linear correlation with the saving rate changes, in which the saving rate change in Inner Mongolia has the largest sensitivity (elasticity 0.09). Full results of sensitivity analysis for all the parameters are given in Supplementary Data 11–20.

Furthermore, the dynamic consequences of investment for the technological progress of energy conservation fall beyond the scope of this study. Studying the dynamic impacts of technological progress on $CO_2$ mitigation due to saving rate changes is an interesting future research direction[28,29]. In addition, due to the limitations of statistical data, the data for China's inter-regional trade in the MRIO table used in this study are from estimations based on inter-provincial railway transportation data[13,30]. In the future, estimating inter-regional trade with domestic trade surveys can steadily increase the accuracy of the calculations.

## Methods

**Impact mechanism of saving rate changes on CO2 emissions.** In order to understand the mechanism for the impacts of saving rate decrease on $CO_2$ emissions, we first construct a theoretical dynamic model with two final products, two representative firms and one representative household. Product 1 is capital goods, while product 2 is consumption goods. Firm 1 produces product 1 and firm 2 produces product 2. The production functions of the two products are shown in Eqs. (1) and (2):

$$y_{1t} = A_1 k_{1t}{}^\alpha \tag{1}$$

$$y_{2t} = A_2 k_{2t}{}^\beta \tag{2}$$

where $y_{1t}$ and $y_{2t}$ indicate the outputs of products 1 and 2, respectively; $A_1$ and $A_2$ represent the total factor productivity of products 1 and 2, respectively; $k_{1t}$ and $k_{2t}$ denote the capital stocks of firms 1 and 2 at time $t$, respectively; and $\alpha$ and $\beta$ stand for the capital input elasticity of firms 1 and 2, respectively. $CO_2$ emissions are considered as the by-products of the production, as shown in Eqs. (3) and (4):

$$l_{1t} = e_1 y_{1t} \tag{3}$$

$$l_{2t} = e_2 y_{2t} \tag{4}$$

where $l_{1t}$ and $l_{2t}$ represent the $CO_2$ emissions from firms 1 and 2, respectively; and $e_1$ and $e_2$ indicate emission intensities ($CO_2$ emissions per unitary output) of firms 1 and 2, respectively.

The representative household is both a capital holder and a firm owner. Therefore, the income of the representative household is calculated by Eq. (5):

$$\iota = y_{1t} + p y_{2t} \tag{5}$$

where $p$ is the price of the consumption goods, and the price of the capital goods is standardized to one.

Then we have the Eq. (6), with $s$ as the saving rate.

$$y_{1t} = s(y_{1t} + p y_{2t}) \tag{6}$$

Because capital goods are used for investment, the left side of Eq. (6) mean the investment. The right side of Eq. (6) represents the savings.

The dynamics of capital stocks are given by Eq. (7):

$$\dot{k}_{1t} + \dot{k}_{2t} = y_{1t} - \delta(k_{1t} + k_{2t}) \tag{7}$$

where the dot · at the top means the time derivative; and $\delta$ is the depreciation rate.

The profit maximization of both firms gives the first-order conditions, as shown in Eqs. (8) and (9):

$$r_t = \alpha A_1 k_{1t}{}^{\alpha-1} \tag{8}$$

$$r_t = p \beta A_2 k_{2t}{}^{\beta-1} \tag{9}$$

where $r_t$ indicates the interest rate. Eqs. (1), (2) and (6) give Eq. (10).

$$(1-s)A_1 k_{1t}{}^\alpha = sp A_2 k_{2t}{}^\beta \tag{10}$$

Equations (8) and (9) give Eq. (11).

$$\alpha A_1 k_{1t}{}^{\alpha-1} = p \beta A_2 k_{2t}{}^{\beta-1} \tag{11}$$

Then, combining Eqs. (10) and (11) provides Eq. (12).

$$k_{2t} = \frac{(1-s)\beta}{s\alpha} k_{1t} \tag{12}$$

Equation (12) shows that the proportion of capital stock of firm 1 to that of firm 2 decreases with the increase of saving rates. It implies that the economy allocates fewer primary resources to the production of capital goods if the saving rates

decrease. By substituting Eq. (12) into Eq. (7), we can derive Eq. (13).

$$\dot{k}_{1t} = \frac{s\alpha}{s\alpha + (1-s)\beta} A_1 k_{1t}{}^\alpha - \delta k_{1t} \tag{13}$$

Finally, the steady states of capital stocks of firms 1 and 2 are given by Eqs. (14) and (15), respectively.

$$k_1^* = \left\{ \frac{s\alpha}{\delta[s\alpha + (1-s)\beta]} A_1 \right\}^{\frac{1}{1-\alpha}} \tag{14}$$

$$k_2^* = \left\{ \frac{s\alpha}{\delta[s\alpha + (1-s)\beta]} A_1 \right\}^{\frac{1}{1-\alpha}} \frac{(1-s)\beta}{s\alpha} \tag{15}$$

From Eq. (14), we witness that, in the steady state, the decrease in saving rates leads to a decrease in the capital stock of firm 1 that produces capital goods. This is due to two reasons. First, the lower demand for capital goods reduces the incentive to accumulate capital stock for firm 1. Second, a lower supply of capital goods reduces the speed of capital accumulation. The direction of the change in the capital stock of firm 2, which produces consumption goods, is unclear, because two different mechanisms have opposite effects. First, the higher demand for consumption goods reduces the incentive to accumulate capital stock for firm 2. Second, a lower supply of capital goods reduces the speed of capital accumulation for the whole economy.

Based on Eqs. (1)–(4), (14) and (15), we can derive Eq. (16) to calculate the total $CO_2$ emissions of the economy ($L$):

$$L = e_1 y_1^* + e_2 y_2^* \tag{16}$$

where $y_1^*$ and $y_2^*$ indicate the outputs of firms 1 and 2 in the steady state, respectively, as shown in Eqs. (17) and (18).

$$y_1^* = A_1 \left\{ \frac{s\alpha}{\delta[s\alpha + (1-s)\beta]} A_1 \right\}^{\frac{\alpha}{1-\alpha}} \tag{17}$$

$$y_2^* = A_2 \left\{ \frac{s\alpha}{\delta[s\alpha + (1-s)\beta]} A_1 \right\}^{\frac{\beta}{1-\alpha}} \left[ \frac{(1-s)\beta}{s\alpha} \right]^\beta \tag{18}$$

The partial derivative of Eq. (16) with respect to $s$ is shown in Eq. (19):

$$L'(s) = e_1 y_1^* \alpha \Omega - e_2 y_2^* \beta \left[ \frac{1}{s(1-s)} - \Omega \right] \tag{19}$$

where $\Omega = \frac{1}{1-\alpha} \left\{ \frac{1}{s} - \frac{\alpha-\beta}{\delta[s\alpha + (1-s)\beta]} \right\}$. Therefore, if the ratio between the emission intensity of the firm producing capital goods and that of the firm producing consumption goods is large enough $\left( \frac{e_1}{e_2} > \frac{y_2^* \beta \left[ \frac{1}{s(1-s)} - \Omega \right]}{y_1^* \alpha \Omega} \right)$, the decrease of the saving rate would reduce the $CO_2$ emissions from the economy. In other words, if the cumulative $CO_2$ emission intensity of the production process of capital goods is much larger than that of the consumption goods, the saving rate decrease would lead to $CO_2$ emission reduction.

**Environmentally extended multi-regional input–output model.** We use a global environmentally extended multi-regional input–output (EE-MRIO) model to evaluate global $CO_2$ emissions in the scenarios. EE-MRIO models have been widely used in the analyses of environmental and social issues[31], including greenhouse gas emissions[13,23,32,33], mercury emissions[34–36], resource scarcity[37], forestry resources[38] and health impacts[39–41]. We construct a global EE-MRIO model by treating global $CO_2$ emissions as the satellite account of the global MRIO table. The global MRIO table describes the inter-regional and intra-regional trade of goods and services at the sectoral level[42]. We use the Global Trade Analysis Project (GTAP) MRIO data to construct the EE-MRIO model because the GTAP database has relatively high resolutions of nations and comparable sectors. GTAP database has a quality control process to prioritize the usage of data sources with higher degrees of reliability[43]. The comparison studies for MRIO databases show that GTAP produces similar results with Eora[44] and WIOD[45] for the majority of regions. Meanwhile, GTAP also has a policy of placing a premium on the continuity of data suppliers[43]. This property makes it suitable for the structural decomposition analysis using two GTAP MRIO tables. The global MRIO table and $CO_2$ emissions were constructed in the previous study, which integrates the Chinese MRIO table with the GTAP MRIO data[13]. The global MRIO table in this study has 57 sectors for 139 nations other than China and 30 sectors for 30 provinces in China (full list of nations, Chinese regions, and sector classifications in Supplementary Data 21 and 22).

**Structural decomposition analysis for saving rate changes.** We use the SDA to investigate the relative contribution of China's saving rate changes to global $CO_2$ emissions during 2007–2012. In this study, we decompose the changes of global $CO_2$ emissions into the changes of the $CO_2$ emission intensity, production input structure, disposable income level of China, saving rates of China and other factors.

First, $CO_2$ emissions from the global production systems are expressed with the global EE-MRIO model, as shown in Eq. (20):

$$\mathbf{e} = \hat{\mathbf{q}} \mathbf{L} \mathbf{f} \tag{20}$$

where $\mathbf{e}$ denotes $CO_2$ emissions from the global production systems; $\mathbf{q}$ is a vector of $CO_2$ emissions per unitary output of each sector, also known as the $CO_2$ emission intensity of sectors; the hat notation $\wedge$ denotes the diagonalization of a vector; $\mathbf{L}$ denotes the Leontief inverse matrix, whose element $l_{ij}$ represents both the direct and indirect inputs from sector $i$ required to satisfy a unit of final demand of products from sector $j$; and $\mathbf{f}$ is a vector of the final demand.

The changes in global $CO_2$ emissions between time 0 and time 1 can be expressed as Eq. (21).

$$\Delta\mathbf{e} = \frac{1}{2}\Delta\hat{\mathbf{q}}\left(\mathbf{L}^0\mathbf{f}^0 + \mathbf{L}^1\mathbf{f}^1\right) + \frac{1}{2}\left(\hat{\mathbf{q}}^1\Delta\mathbf{L}\mathbf{f}^0 + \hat{\mathbf{q}}^0\Delta\mathbf{L}\mathbf{f}^1\right) + \frac{1}{2}\left(\hat{\mathbf{q}}^0\mathbf{L}^0 + \hat{\mathbf{q}}^1\mathbf{L}^1\right)\Delta\mathbf{f} \quad (21)$$

We further decompose $\mathbf{f}$ into the disposable income of China, the proportions of final consumption and capital formation of China, the structures of final consumption and capital formation of China, and other factors (the remaining in the final demand, including inventory changes in China and the final demand of other nations), as shown in Eq. (22):

$$\mathbf{f} = Wp_c\mathbf{s_c} + Wp_i\mathbf{s_i} + \mathbf{r} \quad (22)$$

where $W$ represents the disposable income of China; $p_c$ and $p_i$ denote the proportions of final consumption and capital formation of China, respectively; vectors $\mathbf{s_c}$ and $\mathbf{s_i}$ indicate the structures of final consumption and capital formation of China, respectively; and the vector $\mathbf{r}$ represents the other factors.

Changes in $\mathbf{f}$ can be evaluated with Eq. (23).

$$\Delta\mathbf{f} = \frac{1}{2}\Delta W\left(p_c^0\mathbf{s_c^0} + p_c^1\mathbf{s_c^1}\right) + \frac{1}{2}\left(W^1\Delta p_c\mathbf{s_c^0} + W^0\Delta p_c\mathbf{s_c^1}\right) + \frac{1}{2}\left(W^0p_c^0 + W^1p_c^1\right)\Delta\mathbf{s_c}$$
$$+ \frac{1}{2}\Delta W\left(p_i^0\mathbf{s_i^0} + p_i^1\mathbf{s_i^1}\right) + \frac{1}{2}\left(W^1\Delta p_i\mathbf{s_i^0} + W^0\Delta p_i\mathbf{s_i^1}\right)$$
$$+ \frac{1}{2}\left(W^0p_i^0 + W^1p_i^1\right)\Delta\mathbf{s_i} + \Delta\mathbf{r}$$
$$(23)$$

Consequently, the relative contributions of the decomposed factors are calculated by Eqs. (24)–(31).

The relative contribution of $CO_2$ emission intensity change $\Delta\mathbf{e_q}$:

$$\Delta\mathbf{e_q} = \frac{1}{2}\Delta\hat{\mathbf{q}}\left[\mathbf{L}^0\left(W^0p_c^0\mathbf{s_c^0} + W^0p_i^0\mathbf{s_i^0} + \mathbf{r}^0\right) + \mathbf{L}^1\left(W^1p_c^1\mathbf{s_c^1} + W^1p_i^1\mathbf{s_i^1} + \mathbf{r}^1\right)\right] \quad (24)$$

The relative contribution of production input structure change $\Delta\mathbf{e_L}$:

$$\Delta\mathbf{e_L} = \frac{1}{2}\left[\hat{\mathbf{q}}^1\Delta\mathbf{L}\left(W^0p_c^0\mathbf{s_c^0} + W^0p_i^0\mathbf{s_i^0} + \mathbf{r}^0\right) + \hat{\mathbf{q}}^0\Delta\mathbf{L}\left(W^1p_c^1\mathbf{s_c^1} + W^1p_i^1\mathbf{s_i^1} + \mathbf{r}^1\right)\right]$$
$$(25)$$

The relative contribution of the change in disposable income of China $\Delta\mathbf{e_W}$:

$$\Delta\mathbf{e_W} = \frac{1}{4}\left(\hat{\mathbf{q}}^0\mathbf{L}^0 + \hat{\mathbf{q}}^1\mathbf{L}^1\right)\Delta W(p_c^0\mathbf{s_c^0} + p_i^0\mathbf{s_i^0} + p_c^1\mathbf{s_c^1} + p_i^1\mathbf{s_i^1}) \quad (26)$$

The relative contribution of the change in final consumption proportion in China $\Delta\mathbf{e_{pc}}$:

$$\Delta\mathbf{e_{pc}} = \frac{1}{4}\left(\hat{\mathbf{q}}^0\mathbf{L}^0 + \hat{\mathbf{q}}^1\mathbf{L}^1\right)\left(W^1\Delta p_c\mathbf{s_c^0} + W^0\Delta p_c\mathbf{s_c^1}\right) \quad (27)$$

The relative contribution of the change in final consumption structure in China $\Delta\mathbf{e_{sc}}$:

$$\Delta\mathbf{e_{sc}} = \frac{1}{4}\left(\hat{\mathbf{q}}^0\mathbf{L}^0 + \hat{\mathbf{q}}^1\mathbf{L}^1\right)\left(W^0p_c^0 + W^1p_c^1\right)\Delta\mathbf{s_c} \quad (28)$$

The relative contribution of the change in capital formation proportion in China $\Delta\mathbf{e_{pi}}$:

$$\Delta\mathbf{e_{pi}} = \frac{1}{4}\left(\hat{\mathbf{q}}^0\mathbf{L}^0 + \hat{\mathbf{q}}^1\mathbf{L}^1\right)\left(W^1\Delta p_i\mathbf{s_i^0} + W^0\Delta p_i\mathbf{s_i^1}\right) \quad (29)$$

The relative contribution of the change in capital formation structure in China $\Delta\mathbf{e_{si}}$:

$$\Delta\mathbf{e_{si}} = \frac{1}{4}\left(\hat{\mathbf{q}}^0\mathbf{L}^0 + \hat{\mathbf{q}}^1\mathbf{L}^1\right)\left(W^0p_i^0 + W^1p_i^1\right)\Delta\mathbf{s_i} \quad (30)$$

The relative contribution of the change in other factors $\Delta\mathbf{e_r}$:

$$\Delta\mathbf{e_r} = \frac{1}{2}\left(\hat{\mathbf{q}}^0\mathbf{L}^0 + \hat{\mathbf{q}}^1\mathbf{L}^1\right)\Delta\mathbf{r} \quad (31)$$

This study defines (GDP-consumption)/GDP as the saving rate. Thus, saving and investment rates are two sides of a coin in the framework of this method. By summing up the relative contributions of proportions of final consumption and capital formation in China, we calculate the relative contribution of China's saving rate changes to global $CO_2$ emission changes ($\Delta\mathbf{e_{sr}}$), as shown in Eq. (32).

$$\Delta\mathbf{e_{sr}} = \frac{1}{4}\left(\hat{\mathbf{q}}^0\mathbf{L}^0 + \hat{\mathbf{q}}^1\mathbf{L}^1\right)\left(W^1\Delta p_c\mathbf{s_c^0} + W^0\Delta p_c\mathbf{s_c^1} + W^1\Delta p_i\mathbf{s_i^0} + W^0\Delta p_i\mathbf{s_i^1}\right) \quad (32)$$

**Counterfactual analysis of saving rate decrease in China**. To evaluate the partial effects of the decrease in China's saving rates, this study assumes that global supply chains, sectoral $CO_2$ emission intensity and income levels remain constant, but the saving rate of China changes. We calculate the final demand with changed saving rates with Eq. (33).

$$\mathbf{f}_m = \mathbf{c}_m + \mathbf{i}_m \quad (33)$$

The vector $\mathbf{f}_m$ is the final demand of region $m$; and vectors $\mathbf{c}_m$ and $\mathbf{i}_m$ refer to the final consumption and capital formation of region $m$, respectively.

We define the saving rate $r_m$ as the proportion for the sum of vector $\mathbf{i}_m$ in the sum of vector $\mathbf{f}_m$. If the saving rate of region $m$ increases by $\alpha$ percentage points, then the updated capital formation $\mathbf{i}_m^*$ is given by $\mathbf{i}_m^* = (r_m + \alpha) \times \mathbf{f}_m$. Under the assumption that the sum of the final demand of all the sectors for each Chinese region remains constant, the updated final consumption $\mathbf{c}_m^*$ is calculated as $\mathbf{c}_m^* = \mathbf{c}_m + \mathbf{i}_m - \mathbf{i}_m^*$. We assume that the structures of the final demand and capital formation of each region remain constant. Thus, the updated final consumption and capital formation of sectors are calculated with Eqs. (34) and (35), respectively:

$$c_{m,j}^* = \frac{c_{m,j}}{\sum_j c_{m,j}} \times \sum_j c_{m,j}^* \quad (34)$$

$$i_{m,j}^* = \frac{i_{m,j}}{\sum_j i_{m,j}} \times \sum_j i_{m,j}^* \quad (35)$$

where $c_{m,j}^*$ and $i_{m,j}^*$ are the updated final consumption and capital formation of sector $j$ in region $m$.

Consequently, the updated final demand is given in Eq. (36):

$$\mathbf{f}_m^* = \mathbf{c}_m^* + \mathbf{i}_m^* \quad (36)$$

where $\mathbf{f}_m^*$ is the updated final demand of region $m$.

We estimate global $CO_2$ emissions with changed saving rates using the global EE-MRIO model. The fundamental balance relationship of the global MRIO model is shown in Eq. (37):

$$\mathbf{x}^* = (\mathbf{I} - \mathbf{A})^{-1}\mathbf{f}^* \quad (37)$$

where $\mathbf{x}^*$ is a vector of the updated sectoral total outputs; $\mathbf{I}$ is an identity matrix; $\mathbf{A}$ is the direct input coefficient matrix, whose element $a_{ij}$ represents the direct input from sector $i$ required to produce a unit of output in sector $j$; $(\mathbf{I} - \mathbf{A})^{-1}$ denotes the Leontief inverse matrix, whose element $l_{ij}$ represents both the direct and indirect inputs from sector $i$ required to satisfy a unit of final demand of products from sector $j$; and $\mathbf{f}^*$ is a vector of the updated final demand.

$CO_2$ emissions from the global production systems $\mathbf{e}^{p,*}$ are calculated by Eq. (38):

$$\mathbf{e}^{p,*} = \hat{\mathbf{q}}(\mathbf{I} - \mathbf{A})^{-1}\mathbf{f}^* \quad (38)$$

where $\mathbf{q}$ is a vector of $CO_2$ emissions per unitary output of each sector, also known as the $CO_2$ emission intensity of sectors. The hat notation $\wedge$ denotes the diagonalization of a vector.

For $CO_2$ emissions from Chinese households, since the energy consumption structure of households remains constant, we assume that the $CO_2$ emissions per unit of final consumption remain constant. Therefore, the updated $CO_2$ emissions from Chinese households $e^{h,*}$ are estimated by Eq. (39):

$$e_m^{h,*} = \frac{\sum_j c_{m,j}^*}{\sum_j c_{m,j}} \times e_m^h \quad (39)$$

where $e_m^h$ denotes the $CO_2$ emissions from the households of Chinese region $m$ in 2012, while $c_{m,j}$ and $c_{m,j}^*$ denote the original and updated final consumption of sector $j$ in Chinese region $m$, respectively.

To evaluate the uncertainty of this study, we calculate the range of the $CO_2$ emission changes with the forecasted range of potential saving rate changes in China. It has been forecasted that the saving rates of China will continuously decrease from 2019 to 2035, thus the upper range of saving rate changes is calculated with the forecasted lowest possible saving rate of China during this period[9]. For the lower range, we assume that the saving rate of China will not be higher than the lowest saving rate in the history after the complete reform of market economy in China (from 1994 to 2019)[9].

We use the matrix-based method considering all the parameters of the model developed in this study to investigate the sensitivity of this model[46]. We first calculate the sensitivity coefficient as the change of the $CO_2$ emissions caused by a marginal change in each of the parameters. The sensitivity coefficients of this method are equivalent to the coefficients on the variance of the independent parameters in the error-propagation method[26,47], which show the partial effects of the changes in parameters on the changes in results. For global industrial $CO_2$ emissions, the sensitivity coefficients for the $CO_2$ emission intensity of nation sector $j$, each element $T_{ij}$ in the intermediate transaction matrix $\mathbf{T}$ of the MRIO table, the disposable income ($W_m$) of Chinese region $m$, the consumption structure ($\mathbf{s}_{c,m}$) of Chinese region $m$, the capital formation structure ($\mathbf{s}_{i,m}$) of Chinese region $m$, and the saving rate change $\alpha$, are calculated with Eqs. (40)–(45), respectively. The notation $\Delta e^p$ represents the changes of global industrial $CO_2$ emissions due to the changes in saving rates of China; $s_{c,mj}$ and $s_{i,mj}$ represent the proportions of nation sector $j$ in the final demand and capital formation of region $m$, respectively;

and $x_j$ indicates the total output of nation sector $j$.

$$\frac{\partial \Delta e^p}{\partial q_j} = [\mathbf{L}(\mathbf{f}^* - \mathbf{f})]_j \tag{40}$$

$$\frac{\partial \Delta e^P}{\partial T_{ij}} = \frac{(\mathbf{qL})_i [\mathbf{L}(\mathbf{f}^* - \mathbf{f})]_j}{x_j} \tag{41}$$

$$\frac{\partial \Delta e^P}{\partial W_m} = \mathbf{qL}\alpha(\mathbf{s_{i,m}} - \mathbf{s_{c,m}}) \tag{42}$$

$$\frac{\partial \Delta e^P}{\partial s_{i,mj}} = (\mathbf{qL}W_m\alpha)_j \tag{43}$$

$$\frac{\partial \Delta e^P}{\partial s_{c,mj}} = -(\mathbf{qL}W_m\alpha)_j \tag{44}$$

$$\frac{\partial \Delta e^P}{\partial \alpha} = \sum_m [\mathbf{qL}W_m(\mathbf{s_{i,m}} - \mathbf{s_{c,m}})] \tag{45}$$

For $CO_2$ emissions from Chinese households, the sensitivity coefficient for the saving rate change $\alpha$ is calculated by Eq. (46):

$$\frac{\partial \Delta e_m^h}{\partial \alpha} = -\left(\frac{e_m^h}{1 - r_m}\right) \tag{46}$$

where $\Delta e_m^h$ represents the change of $CO_2$ emissions from households in Chinese region $m$ due to the changes in saving rates of China, and $r_m$ indicates the current saving rate of Chinese region $m$.

To further eliminate the effect caused by the statistical units of the parameters, we define the dimensionless elasticities of the parameters, which indicate the ratios between the changing rate of global $CO_2$ emission changes and the changing rate of the parameters. The elasticities are calculated by Eqs. (47)–(53):

$$EL_{q_j}^P = \frac{\partial \Delta e^P}{\partial q_j} \times \frac{q_j}{\Delta e^P} \tag{47}$$

$$EL_{T_{ij}}^P = \frac{\partial \Delta e^P}{\partial T_{ij}} \times \frac{T_{ij}}{\Delta e^P} \tag{48}$$

$$EL_{W_m}^P = \frac{\partial \Delta e^P}{\partial W_m} \times \frac{W_m}{\Delta e^P} \tag{49}$$

$$EL_{s_{c,mj}}^P = \frac{\partial \Delta e^P}{\partial s_{c,mj}} \times \frac{s_{c,mj}}{\Delta e^P} \tag{50}$$

$$EL_{s_{i,mj}}^P = \frac{\partial \Delta e^P}{\partial s_{i,mj}} \times \frac{s_{i,mj}}{\Delta e^P} \tag{51}$$

$$EL_\alpha^P = \frac{\partial \Delta e^P}{\partial \alpha} \times \frac{\alpha}{\Delta e^P} \tag{52}$$

$$EL_\alpha^h = \frac{\partial \Delta e^h}{\partial \alpha} \times \frac{\alpha}{\Delta e^h} \tag{53}$$

where $EL_{q_j}^P$, $EL_{T_{ij}}^P$, $EL_{W_m}^P$, $EL_{s_{c,mj}}^P$, $EL_{s_{i,mj}}^P$ and $EL_\alpha^P$ represent the elasticities for global industrial $CO_2$ emissions to the $CO_2$ emission intensity, the elements of intermediate transaction matrix, the disposal incomes of Chinese regions, the final consumption structure of Chinese regions, the capital formation structure of Chinese regions and the saving rate changes, respectively. The notation $EL_\alpha^h$ denotes the elasticities for the $CO_2$ emissions of Chinese households to the saving rate changes.

To evaluate the global $CO_2$ emissions induced by a unit basket of final consumption or capital formation in China, we first sum the final consumption columns and capital formation columns of Chinese regions to form vectors of the final consumption and capital formation of China. Then, we define vectors representing a unit basket of final consumption ($\mathbf{c^u}$) and a unit basket of capital formation ($\mathbf{i^u}$) in China by Eqs. (54) and (55), respectively:

$$c_j^u = \frac{c_j^*}{\sum_j c_j^*} \tag{54}$$

$$i_j^u = \frac{i_j^*}{\sum_j i_j^*} \tag{55}$$

where $c_j^u$ and $i_j^u$ denote the proportions of sector $j$ in the final consumption and capital formation of China, respectively.

Subsequently, we calculate the global $CO_2$ emissions induced by a unit basket of final consumption ($\mathbf{e^{u,c}}$) and those induced by a unit basket of capital formation

($\mathbf{e^{u,i}}$) in China by diagonalizing $\mathbf{c^u}$ and $\mathbf{i^u}$, as shown in Eqs. (56) and (57), respectively.

$$\mathbf{e^{u,c}} = \mathbf{q}(\mathbf{I} - \mathbf{A})^{-1}\mathbf{\widehat{c}^u} \tag{56}$$

$$\mathbf{e^{u,i}} = \mathbf{q}(\mathbf{I} - \mathbf{A})^{-1}\mathbf{\widehat{i}^u} \tag{57}$$

The hat notation ^ denotes the diagonalization of a vector.

To estimate global $CO_2$ emissions in the "consummate greener consumption" scenario, we assume that the commodities produced by the most efficient nation concerning cumulative (including both direct and indirect) $CO_2$ emission intensity replace the current commodities consumed in Chinese regions.

First, we calculate the vector of the cumulative $CO_2$ emission intensity $\mathbf{q^L}$ with Eq. (58).

$$\mathbf{q^L} = \mathbf{q}(\mathbf{I} - \mathbf{A})^{-1} \tag{58}$$

Second, we select the lowest cumulative $CO_2$ emission intensity for each sector and form a vector $\mathbf{q^{L,ext}}$. We also select the final demands of Chinese regions and sum them as a column vector $\mathbf{f^{chn}}$.

Finally, we can estimate the global $CO_2$ emissions of the "consummate greener consumption" scenario $\mathbf{e^{ext}}$ with Eq. (59):

$$\mathbf{e^{ext}} = \mathbf{q^{L,ext}} \times \mathbf{\widehat{f^{chn}}} \tag{59}$$

Notably, the cumulative $CO_2$ emission intensity of the petroleum and coal products sector in Botswana is extremely low. For example, the second-lowest cumulative $CO_2$ emission intensity is over 362 times larger than that of Botswana, and the global average is over 3400 times larger than that of Botswana. We regard the cumulative $CO_2$ emission intensity of petroleum and coal products in Botswana as abnormal. Thus, we use the second-lowest $CO_2$ emission intensity of this sector as the corresponding basis.

This study constructs four scenarios regarding changes in the saving rates and regional settings (Table 1). For changes in the saving rates, one assumption is the 15-percentage-point decrease, based on the forecast that China's national saving rate is expected to decrease from 47 to 32% during 2019–2035[9]. The other assumption is that the saving rates of all Chinese regions change to the lowest regional saving rate (i.e., Shanghai's saving rate, 37%) in 2012. The historical data show that the changes in the saving and investment rates in China are synchronous in most years[4]. Therefore, in both settings, changes of the investment rates are assumed to be synchronous with changes in the saving rates. For regional settings, this study first measures the changes in global $CO_2$ emissions caused by the simultaneous saving rate changes of all Chinese regions; then it examines the changes in global $CO_2$ emissions caused by the changes in the saving rate of only one Chinese region at a time. In all these scenarios, we assume that the production levels are flexible. This means that the global production structure (represented by the direct input coefficient matrix of the global input–output model) remains constant, and changes in the final consumption and capital formation of Chinese regions lead to changes in global total outputs and subsequent global $CO_2$ emissions. We also explore the situations with fixed production levels, as described in the Supplementary Note 1 and Supplementary Fig. 7. To focus on the static property of the changes in saving rates, this paper does not consider the issues of economic growth. Thus, long-term economic growth is not considered.

**Reporting summary**. Further information on research design is available in the Nature Research Reporting Summary linked to this article.

## Data availability
The data used in the global EE-MRIO model and the estimations of $CO_2$ emissions include the global MRIO table and the sectoral $CO_2$ emissions of nations and Chinese regions. The global MRIO table can be obtained from the previous study[13], while the data for $CO_2$ emissions can be obtained from the previous literature[48,49]. The source data underlying Figs. 1–5, Supplementary Figs. 1–7 are provided as a Source Data file. All datasets generated during this study are available from the corresponding authors upon reasonable request.

## Code availability
All computer codes generated during this study are available from the corresponding authors upon reasonable request.

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

## Acknowledgements

This study was financially supported by the National Natural Science Foundation of China (51721093 and 71874014) and the Fundamental Research Funds for the Central Universities.

## Author contributions

S.L. and C.L. designed this study. J.Q., C.F., Y.L., X.Y., Y.L. and Z.M. collected the data. C.L., J.Q. and C.F. conducted the calculations and interpretations of the results. C.L., J.Q., S.L., C.F., T.W. and Z.Y. wrote and revised the paper. S.L. supervised this study.

## Competing interests

The authors declare no competing interests.
