## [Peer Review File · Nature Communications]

Reviewers' comments:

Reviewer #1 (Remarks to the Author):

Reviewer's report for

Saving less in China facilitates global CO₂ mitigation

Suggested revision: reject and publish in a different journal

General impression:

This paper shows a methodology for calculating the global effects of changing saving rates in China on CO₂ emissions using input-output analysis. The methodology and calculations in this paper are sound and the message of this paper is clear. The paper is well-written and well-prepared and – in my opinion – does not require much work for publication. I do however think that this paper should not be published in Nature Communications.

Justification of opinion:

The authors commence using an assumption that the savings rate in China is correlated to the investment rate, which in return is correlated to the consumption required for these investments. I cannot comment on whether this assumption is correct, but the authors justify it very well and I will therefore accept it.

Using the savings rate as a proxy, the authors assess how a change in the savings rate will change the consumption, and as a result of this change, how the CO₂ footprint around the world changes.

In my opinion, it is unrealistic to assume that only the savings rate changes and all other effects that might contribute to a change in consumption can be ignored. In my view, this is a theoretical case.

Once the change in consumption has been determined, the calculation of the corresponding change in CO₂ footprints around the world using a nested sub-national/global input-output framework is a standard calculation that has been widely used in the literature over the last 15 years.

The results show that the changes in CO₂ emissions around the world can be quite small for individual countries. Given that all input-output databases are subject to certain margins, I believe that a thorough sensitivity analysis (which is not presented in this study) will show that a – for example – the increased emissions in Australia of 1.8Mt (which is approx 0.5% of Australia's territorial annual emissions) might as well be a negative number. The same holds for other numbers, which are often relatively small compared to the nations' total emissions.

Hence, while the calculations are theoretically correct, I suspect that the claims that the authors make regarding the savings/increase of emissions in partnering countries around the world need to be backed up by robust sensitivity analysis results.

Assuming that the input-output data were 100% accurate the results therefore fully reliable, this study would show that a decrease in consumption in China will lead to a decreased footprint around the world – split up into the contribution of the individual countries in China's supply chain network. I do not see the scientific contribution of this result to justify a publication in Nature Communications – especially given that the change in consumption is

derived from the isolated effect of a changing savings rate (see above). Global footprints, especially carbon footprints have been published in papers for more than a decade now (Alsamawi, Murray, & Lenzen, 2014; Alsamawi, Murray, Lenzen, Moran, & Kanemoto, 2014; Hertwich & Peters, 2009; Liang et al., 2016; Moran, McBain, Kanemoto, Lenzen, & Geschke, 2015; Tom, Fischbeck, & Hendrickson, 2015; Wiedmann et al., 2015; Xiao et al., 2017)

References

- Alsamawi, A., Murray, J., & Lenzen, M. (2014). The Employment Footprints of Nations. *Journal of Industrial Ecology*, *18*(1), 59-70. doi:10.1111/jiec.12104
- Alsamawi, A., Murray, J., Lenzen, M., Moran, D., & Kanemoto, K. (2014). The Inequality Footprints of Nations: A Novel Approach to Quantitative Accounting of Income Inequality. *PLOS ONE*, *9*(10), e110881. doi:10.1371/journal.pone.0110881
- Hertwich, E. G., & Peters, G. P. (2009). Carbon Footprint of Nations: A Global, Trade-Linked Analysis. *Environmental Science & Technology*, *43*(16), 6414-6420. doi:10.1021/es803496a
- Liang, S., Guo, S., Newell, J. P., Qu, S., Feng, Y., Chiu, A. S. F., & Xu, M. (2016). Global Drivers of Russian Timber Harvest. *Journal of Industrial Ecology*, *20*(3), 515-525. doi:10.1111/jiec.12417
- Moran, D., McBain, D., Kanemoto, K., Lenzen, M., & Geschke, A. (2015). Global Supply Chains of Coltan. *Journal of Industrial Ecology*, *19*(3), 357-365. doi:10.1111/jiec.12206
- Tom, M., Fischbeck, P., & Hendrickson, C. (2015). Energy use, blue water footprint, and greenhouse gas emissions for current food consumption patterns and dietary recommendations in the US. *Environment Systems and Decisions*, 1-12. doi:10.1007/s10669-015-9577-y
- Wiedmann, T. O., Schandl, H., Lenzen, M., Moran, D., Suh, S., West, J., & Kanemoto, K. (2015). The material footprint of nations. *Proceedings of the National Academy of Sciences*, *112*(20), 6271-6276. doi:10.1073/pnas.1220362110
- Xiao, Y., Lenzen, M., Benoît-Norris, C., Norris, G. A., Murray, J., & Malik, A. (2017). The Corruption Footprints of Nations. *Journal of Industrial Ecology*, n/a-n/a. doi:10.1111/jiec.12537

Reviewer #2 (Remarks to the Author):

Thank you very much for sending this interesting piece of work. Although the implications China's booming economy on global CO2 emissions are well-known, the influence of China's saving rates was not quantified previously.

Overall, the article is well-written and the analysis presented in the study is convincing. However, I have a few concerns about the paper, which could be addressed.

Major comment:

The study does not address the dynamics in the economic systems of the world as it only focuses only on one aspect of the economy. Although this limitation is described in the article, the assumption has severe implications on the final outcomes of the study and therefore, on policy-making. Hence, it would be interesting to undertake an uncertainty analysis to show how other factors could have altered the findings of the study.

Minor comments:

1. Is there a specific reason to choose the GTAP database? Because there are alternatives such as the Eora database (which also includes the regional MRIO database)
2. The recommendations provided in the Discussion section could have been supported with relevant references.
3. There are a few typos, which requires a spell-check. For example, the United States of American (USA) on page 8

Kind regards,

Chanjief Chandrakumar, PhD

Responses to reviewers' comments

Reviewer: 1

This paper shows a methodology for calculating the global effects of changing saving rates in China on CO₂ emissions using input-output analysis. The methodology and calculations in this paper are sound and the message of this paper is clear. The paper is well-written and well-prepared and – in my opinion – does not require much work for publication. I do however think that this paper should not be published in Nature Communications.

Answer:

Thank you for your comments. We have revised our manuscript according to your comments. Details are shown in our responses to your following comments. We hope this revised manuscript can address your concerns.

1. The authors commence using an assumption that the savings rate in China is correlated to the investment rate, which in return is correlated to the consumption required for these investments. I cannot comment on whether this assumption is correct, but the authors justify it very well and I will therefore accept it.

Using the savings rate as a proxy, the authors assess how a change in the savings rate will change the consumption, and as a result of this change, how the CO₂ footprint around the world changes.

In my opinion, it is unrealistic to assume that only the savings rate changes and all other effects that might contribute to a change in consumption can be ignored. In my view, this is a theoretical case.

Once the change in consumption has been determined, the calculation of the corresponding change in CO₂ footprints around the world using a nested sub-national/global input-output framework is a standard calculation that has been widely used in the literature over the last 15 years.

Answer:

Thank you for your comment.

This study aims to reveal the potential global CO₂ emission changes due to the foreseeable saving rate decrease in China, which would reduce the capital formation and increase the final consumption of China. Although the factors other than the change of saving rates also affect the emissions, in the revised manuscript, we extract the effect of the change of saving rates by using both theoretical and empirical methods. In the theoretical dynamic model, we reveal the mechanism for the impacts of saving rate decrease on CO₂ emissions when the other factors, such as capital dynamics and input substitution, are considered. In the empirical part, we use a decomposition method to extract the relative contribution of China's saving rate changes when other

factors, such as the technology progress, input substitution, and the change of consumption structure due to other reasons, are controlled by a quantitative method. We have added a paragraph with the mechanism for the impacts of saving rate changes on CO₂ emissions, to clarify this point, as shown in lines 392-472:

“We first propose a theoretical dynamic model to reveal the mechanism for the impacts of saving rate decrease on CO₂ emissions. We then use a newly proposed multi-sectoral decomposition method to quantitatively extract the relative contribution of China’s saving rate changes to global CO₂ emissions.

The mechanism for the impacts of saving rate changes on CO₂ emissions

In order to understand the mechanism for the impacts of saving rate decrease on CO₂ emissions, we first construct a theoretical dynamic model with two final products, two representative firms, and one representative household. Product 1 is capital goods, while product 2 is consumption goods. Firm 1 produces product 1, and firm 2 produces product 2. The production functions of the two products are shown in equations (1) and (2):

$$y_{1t} = A_1 k_{1t}^\alpha \quad (1)$$

$$y_{2t} = A_2 k_{2t}^\beta \quad (2)$$

where y_{1t} and y_{2t} indicate the outputs of products 1 and 2, respectively; A_1 and A_2 represent the total factor productivity of products 1 and 2, respectively; k_{1t} and k_{2t} denote the capital stocks of firms 1 and 2 at time t , respectively; and α and β stand for the capital input elasticity of firms 1 and 2, respectively. CO₂ emissions are considered as the by-products of the production, as shown in equations (3) and (4):

$$l_{1t} = e_1 y_{1t} \quad (3)$$

$$l_{2t} = e_2 y_{2t} \quad (4)$$

where l_{1t} and l_{2t} represent the CO₂ emissions from firms 1 and 2, respectively; and e_1 and e_2 indicate emission intensities (CO₂ emissions per unitary output) of firms 1 and 2, respectively.

The representative household is both a capital holder and a firm owner. Therefore, the income of the representative household is calculated by equation (5):

$$t = y_{1t} + p y_{2t} \quad (5)$$

where p is the price of the consumption goods, and the price of the capital goods is standardized to one.

Then we have the equation (6), with s as the saving rate.

$$y_{1t} = s(y_{1t} + p y_{2t}) \quad (6)$$

Because capital goods are used for investment, the left side of equation (6) mean the investment. The right side of equation (6) represents the savings.

The dynamics of capital stocks are given by equation (7):

$$\dot{k}_{1t} + \dot{k}_{2t} = y_{1t} - \delta(k_{1t} + k_{2t}) \quad (7)$$

where the dot $\dot{\cdot}$ at the top means the time derivative; and δ is the depreciation rate.

The profit maximization of both firms gives the first-order conditions, as shown in equations (8) and (9):

$$r_t = \alpha A_1 k_{1t}^{\alpha-1} \quad (8)$$

$$r_t = p\beta A_2 k_{2t}^{\beta-1} \quad (9)$$

where r_t indicates the interest rate. Equations (1), (2), and (6) give equation (10).

$$(1-s)A_1 k_{1t}^\alpha = spA_2 k_{2t}^\beta \quad (10)$$

Equations (8) and (9) give equation (11).

$$\alpha A_1 k_{1t}^{\alpha-1} = p\beta A_2 k_{2t}^{\beta-1} \quad (11)$$

Then, combining equations (10) and (11) provides equation (12).

$$k_{2t} = \frac{(1-s)\beta}{s\alpha} k_{1t} \quad (12)$$

Equation (12) shows that the proportion of capital stock of firm 1 to that of firm 2 decreases with the increase of saving rates. It implies that the economy allocates fewer primary resources to the production of capital goods if the saving rates decrease. By substituting equation (12) into equation (7), we can derive equation (13).

$$\dot{k}_{1t} = \frac{s\alpha}{s\alpha+(1-s)\beta} A_1 k_{1t}^\alpha - \delta k_{1t} \quad (13)$$

Finally, the steady states of capital stocks of firms 1 and 2 are given by equations (14) and (15), respectively.

$$k_1^* = \left(\frac{s\alpha}{\delta(s\alpha+(1-s)\beta)} A_1 \right)^{\frac{1}{1-\alpha}} \quad (14)$$

$$k_2^* = \left(\frac{s\alpha}{\delta(s\alpha+(1-s)\beta)} A_1 \right)^{\frac{1}{1-\alpha}} \frac{(1-s)\beta}{s\alpha} \quad (15)$$

From equation (14), we witness that, in the steady state, the decrease in saving rates leads to a decrease in the capital stock of firm 1 that produces capital goods. This is due to two reasons. First, the lower demand for capital goods reduces the incentive to accumulate

capital stock for firm 1. Second, a lower supply of capital goods reduces the speed of capital accumulation. The direction of the change in the capital stock of firm 2, which produces consumption goods, is unclear, because two different mechanisms have opposite effects. First, the higher demand for consumption goods reduces the incentive to accumulate capital stock for firm 2. Second, a lower supply of capital goods reduces the speed of capital accumulation for the whole economy.

Based on equations (1), (2), (3), (4), (14), and (15), we can derive equation (16) to calculate the total CO₂ emissions of the economy (L):

$$L = e_1 y_1^* + e_2 y_2^* \quad (16)$$

where y_1^* and y_2^* indicate the outputs of firms 1 and 2 in the steady state, respectively, as shown in equations (17) and (18).

$$y_1^* = A_1 \left(\frac{s\alpha}{\delta(s\alpha + (1-s)\beta)} A_1 \right)^{\frac{\alpha}{1-\alpha}} \quad (17)$$

$$y_2^* = A_2 \left(\frac{s\alpha}{\delta(s\alpha + (1-s)\beta)} A_1 \right)^{\frac{\beta}{1-\alpha}} \left(\frac{(1-s)\beta}{s\alpha} \right)^{\beta} \quad (18)$$

The partial derivative of equation (16) with respect to s is shown in equation (19):

$$L'(s) = e_1 y_1^* \alpha \Omega - e_2 y_2^* \beta \left(\frac{1}{s(1-s)} - \Omega \right) \quad (19)$$

where $\Omega = \frac{1}{1-\alpha} \left(\frac{1}{s} - \frac{\alpha-\beta}{\delta(s\alpha + (1-s)\beta)} \right)$. Therefore, if the ratio between the emission intensity of the firm producing capital goods and that of the firm producing consumption goods is large enough $\left(\frac{e_1}{e_2} > \frac{y_2^* \beta \left(\frac{1}{s(1-s)} - \Omega \right)}{y_1^* \alpha \Omega} \right)$, the decrease of the saving rate would reduce the CO₂ emissions from the economy. In other words, if the cumulative CO₂ emission intensity of the production process of capital goods is much larger than that of the consumption goods, the saving rate decrease would lead to CO₂ emission reduction.”

2. The results show that the changes in CO₂ emissions around the world can be quite small for individual countries. Given that all input-output databases are subject to certain margins, I believe that a thorough sensitivity analysis (which is not presented in this study) will show that a – for example – the increased emissions in Australia of 1.8Mt (which is approx 0.5% of Australia’s territorial annual emissions) might as well be a negative number. The same holds for other numbers, which are often relatively small compared to the nations’ total emissions.

Hence, while the calculations are theoretically correct, I suspect that the claims that the authors make regarding the savings/increase of emissions in partnering countries around the world need to be backed up by robust sensitivity analysis results.

Answer:

Thank you for your comment.

We have added both sensitivity analysis and uncertainty analysis in the revised manuscript. The sensitivity coefficient of this model equals to 1, while the positive/negative sign for CO₂ emission changes of each region would remain constant regardless of the range of saving rate decrease. This can be explained as follows.

In this study, we define the saving rate (r_m) as the proportion of capital formation (i_m) in the final demand (f_m) of region m . If the saving rate of region m changes by α percentage points, the updated capital formation (i_m^*) is given by $i_m^* = (r_m + \alpha) \times f_m$, while the updated final consumption (c_m^*) is calculated as $c_m^* = c_m + i_m - i_m^*$. Furthermore, this model is constructed on the basis that: 1) the global production structure (A matrix of the Leontief input-output model) remains constant; 2) the sum of the final demand of all the sectors for each Chinese region remains constant; and 3) the structures of the final demand and capital formation of each region remain constant. Thus, the CO₂ emission changes of the global production system ($e^{p,*} = \hat{q}(I - A)^{-1}f^*$) and those of Chinese households ($e_m^{h,*} = \frac{\sum_j c_{m,j}^*}{\sum_j c_{m,j}} \times e_m^h$) are in a linear correlation with the changes of saving rates, which makes the sensitivity coefficient of this model equal to 1, and the positive/negative sign for CO₂ emission changes of each region would not change with the range of saving rate changes.

We have revised the manuscript by adding a paragraph with the results of uncertainty analysis and sensitivity analysis, as shown in lines 371-389:

“Uncertainty in the results of this study mainly comes from the uncertainties of the forecasted saving rate changes of China and the global MRIO data. This study estimates the upper and lower ranges of the CO₂ emission changes based on the forecasted range of potential saving rate changes in China. The range for global industrial CO₂ emission reduction is 292.4–513.3 Mt, while the range for CO₂ emissions increments of Chinese households is 124.3–218.2 Mt. Thus, the range of total CO₂ emissions reduction caused by saving rate changes is 168.1–295.1 Mt (more results in Table S10). Improving the accuracy of the forecasting for saving rates in China would reduce the uncertainties of this study. Meanwhile, the data quality of various global MRIO databases varies due to different compiling methods²⁶. Harmonizing the data quality of various global MRIO databases would reduce the uncertainties of this study. Furthermore, the dynamic consequences of investment for the technological progress of energy conservation fall beyond the scope of this study. Studying the dynamic impacts of technological progress on CO₂ mitigation due to saving rate changes is an interesting future research direction^{27,28}. In addition, due to the limitations of statistical data, the data for China’s inter-regional trade in the MRIO table used in this study are from estimations based on inter-provincial railway

transportation data^{13,29}. In the future, estimating inter-regional trade with domestic trade surveys can steadily increase the accuracy of the calculations.”

We also added a paragraph to describe the methods used to evaluate the uncertainties and the sensitivity of the model, as shown in lines 603-615:

“To evaluate the uncertainty of this study, we calculate the range of the CO₂ emission changes with the forecasted range of potential saving rate changes in China. It has been forecasted that the saving rates of China will continuously decrease from 2019 to 2035, thus the upper range of saving rate changes is calculated with the forecasted lowest possible saving rate of China during this period⁹. For the lower range, we assume that the saving rate of China will not be higher than the lowest saving rate in the history after the complete reform of market economy in China (from 1994 to 2019)⁹.

To investigate the sensitivity of this model, we define the sensitivity coefficient s as the ratio between the changing rate of global CO₂ emission reduction and the changing rate of saving rate changes in China. Since the CO₂ emission changes are in a linear correlation with the saving rate changes in the model developed in this study, the sensitivity coefficient of this model equals to 1.”

References:

- 9 Chen, Y., Guo, Y. & Yao, Y. The impact of population aging on high savings rate. *J. Financ. Res.*, 71-84, 2014.
- 13 Mi, Z. et al. Chinese CO₂ emission flows have reversed since the global financial crisis. *Nat. Commun.* 8, 1712, 2017.
- 26 Inomata, S. & Owen, A. Comparative evaluation of MRIO databases. *Econ. Syst. Res.* 26, 239-244, 2014.
- 27 Awaworyi Churchill, S., Inekwe, J., Smyth, R. & Zhang, X. R&D intensity and carbon emissions in the G7: 1870–2014. *Energ. Econ.* 80, 30-37, 2019.
- 28 Jiao, J., Jiang, G. & Yang, R. Impact of R&D technology spillovers on carbon emissions between China’s regions. *Struct. Change Econ. D.* 47, 35-45, 2018.
- 29 Mi, Z. et al. A multi-regional input-output table mapping China's economic outputs and interdependencies in 2012. *Sci. Data* 5, 180155, 2018.

3. Assuming that the input-output data were 100% accurate the results therefore fully reliable, this study would show that a decrease in consumption in China will lead to a decreased footprint around the world – split up into the contribution of the individual countries in China’s supply chain network. I do not see the scientific contribution of this result to justify a publication in Nature Communications – especially given that the change in consumption is derived from the isolated effect of a changing savings rate (see above). Global footprints, especially carbon footprints have been published in papers for more than a decade now

(Alsamawi, Murray, & Lenzen, 2014; Alsamawi, Murray, Lenzen, Moran, & Kanemoto, 2014; Hertwich & Peters, 2009; Liang et al., 2016; Moran, McBain, Kanemoto, Lenzen, & Geschke, 2015; Tom, Fischbeck, & Hendrickson, 2015; Wiedmann et al., 2015; Xiao et al., 2017).

Answer:

Thank you for your comment.

In this study, we assume that the saving rates of China decrease, which means that the shares of the capital formation in the final demand of China decrease and the shares of the final consumption increase. Previous studies, including those you mentioned, have proved that carbon footprint would increase with the increase of consumption. However, results of this study show that, global CO₂ emissions would reduce (rather than increase), even though the final consumption of China increase as the result of the saving rate decrease. The mechanism of global CO₂ emission reduction caused by saving rate decrease is that the cumulative CO₂ emission intensities of sectors producing capital goods are generally larger than those of sectors producing consumption goods (please refer to the response to comment #1). Findings of this study are different from certain studies evaluating the CO₂ emissions of consumption growth in China because this study considers the off-setting effect of final consumption on capital formation, which makes this study insightful for CO₂ mitigation in China, and the implement of the Paris Agreement. These points have been clarified in the Discussion section.

Reviewer: 2

Thank you very much for sending this interesting piece of work. Although the implications China's booming economy on global CO₂ emissions are well-known, the influence of China's saving rates was not quantified previously.

Overall, the article is well-written and the analysis presented in the study is convincing. However, I have a few concerns about the paper, which could be addressed.

Answer:

Thank you for your comments. We have revised our manuscript according to your comments. Details are shown in our responses to your following comments.

Major comment:

The study does not address the dynamics in the economic systems of the world as it only focuses only on one aspect of the economy. Although this limitation is described in the article, the assumption has severe implications on the final outcomes of the study and therefore, on policy-making. Hence, it would be interesting to undertake an uncertainty analysis to show how other factors could have altered the findings of the study.

Answer:

Thank you for your comment.

In this revised manuscript, in addition to our empirical decomposition model, we constructed a theoretical dynamic model to reveal the mechanism for the impacts of saving rate decrease on CO₂ emissions. In the theoretical dynamic model, we show the impacts of saving rate decrease on CO₂ emissions when the other factors, such as capital dynamics and input substitution, are considered. The critical factor for the impacts of saving rate decrease on CO₂ emissions is the different CO₂ emission intensities (the emissions for unitary output) between sectors producing capital goods and those producing consumption goods. This means that the assumption in this study does not have a massive influence on the final outcomes of the study. We have added a paragraph with the mechanism for the impacts of saving rate changes on CO₂ emissions, to clarify this point, as shown in lines 392-472:

“We first propose a theoretical dynamic model to reveal the mechanism for the impacts of saving rate decrease on CO₂ emissions. We then use a newly proposed multi-sectoral decomposition method to quantitatively extract the relative contribution of China’s saving rate changes to global CO₂ emissions.

The mechanism for the impacts of saving rate changes on CO₂ emissions

In order to understand the mechanism for the impacts of saving rate decrease on CO₂ emissions, we first construct a theoretical dynamic model with two final products, two representative firms, and one representative household. Product 1 is capital goods, while

product 2 is consumption goods. Firm 1 produces product 1, and firm 2 produces product 2. The production functions of the two products are shown in equations (1) and (2):

$$y_{1t} = A_1 k_{1t}^\alpha \quad (1)$$

$$y_{2t} = A_2 k_{2t}^\beta \quad (2)$$

where y_{1t} and y_{2t} indicate the outputs of products 1 and 2, respectively; A_1 and A_2 represent the total factor productivity of products 1 and 2, respectively; k_{1t} and k_{2t} denote the capital stocks of firms 1 and 2 at time t , respectively; and α and β stand for the capital input elasticity of firms 1 and 2, respectively. CO_2 emissions are considered as the by-products of the production, as shown in equations (3) and (4):

$$l_{1t} = e_1 y_{1t} \quad (3)$$

$$l_{2t} = e_2 y_{2t} \quad (4)$$

where l_{1t} and l_{2t} represent the CO_2 emissions from firms 1 and 2, respectively; and e_1 and e_2 indicate emission intensities (CO_2 emissions per unitary output) of firms 1 and 2, respectively.

The representative household is both a capital holder and a firm owner. Therefore, the income of the representative household is calculated by equation (5):

$$v = y_{1t} + p y_{2t} \quad (5)$$

where p is the price of the consumption goods, and the price of the capital goods is standardized to one.

Then we have the equation (6), with s as the saving rate.

$$y_{1t} = s(y_{1t} + p y_{2t}) \quad (6)$$

Because capital goods are used for investment, the left side of equation (6) mean the investment. The right side of equation (6) represents the savings.

The dynamics of capital stocks are given by equation (7):

$$\dot{k}_{1t} + \dot{k}_{2t} = y_{1t} - \delta(k_{1t} + k_{2t}) \quad (7)$$

where the dot \cdot at the top means the time derivative; and δ is the depreciation rate.

The profit maximization of both firms gives the first-order conditions, as shown in equations (8) and (9):

$$r_t = \alpha A_1 k_{1t}^{\alpha-1} \quad (8)$$

$$r_t = p \beta A_2 k_{2t}^{\beta-1} \quad (9)$$

where r_t indicates the interest rate. Equations (1), (2), and (6) give equation (10).

$$(1 - s)A_1k_{1t}^\alpha = spA_2k_{2t}^\beta \quad (10)$$

Equations (8) and (9) give equation (11).

$$\alpha A_1k_{1t}^{\alpha-1} = p\beta A_2k_{2t}^{\beta-1} \quad (11)$$

Then, combining equations (10) and (11) provides equation (12).

$$k_{2t} = \frac{(1-s)\beta}{s\alpha} k_{1t} \quad (12)$$

Equation (12) shows that the proportion of capital stock of firm 1 to that of firm 2 decreases with the increase of saving rates. It implies that the economy allocates fewer primary resources to the production of capital goods if the saving rates decrease. By substituting equation (12) into equation (7), we can derive equation (13).

$$\dot{k}_{1t} = \frac{s\alpha}{s\alpha+(1-s)\beta} A_1k_{1t}^\alpha - \delta k_{1t} \quad (13)$$

Finally, the steady states of capital stocks of firms 1 and 2 are given by equations (14) and (15), respectively.

$$k_1^* = \left(\frac{s\alpha}{\delta(s\alpha+(1-s)\beta)} A_1 \right)^{\frac{1}{1-\alpha}} \quad (14)$$

$$k_2^* = \left(\frac{s\alpha}{\delta(s\alpha+(1-s)\beta)} A_1 \right)^{\frac{1}{1-\alpha}} \frac{(1-s)\beta}{s\alpha} \quad (15)$$

From equation (14), we witness that, in the steady state, the decrease in saving rates leads to a decrease in the capital stock of firm 1 that produces capital goods. This is due to two reasons. First, the lower demand for capital goods reduces the incentive to accumulate capital stock for firm 1. Second, a lower supply of capital goods reduces the speed of capital accumulation. The direction of the change in the capital stock of firm 2, which produces consumption goods, is unclear, because two different mechanisms have opposite effects. First, the higher demand for consumption goods reduces the incentive to accumulate capital stock for firm 2. Second, a lower supply of capital goods reduces the speed of capital accumulation for the whole economy.

Based on equations (1), (2), (3), (4), (14), and (15), we can derive equation (16) to calculate the total CO₂ emissions of the economy (L):

$$L = e_1y_1^* + e_2y_2^* \quad (16)$$

where y_1^* and y_2^* indicate the outputs of firms 1 and 2 in the steady state, respectively, as shown in equations (17) and (18).

$$y_1^* = A_1 \left(\frac{s\alpha}{\delta(s\alpha+(1-s)\beta)} A_1 \right)^{\frac{\alpha}{1-\alpha}} \quad (17)$$

$$y_2^* = A_2 \left(\frac{s\alpha}{\delta(s\alpha+(1-s)\beta)} A_1 \right)^{\frac{\beta}{1-\alpha}} \left(\frac{(1-s)\beta}{s\alpha} \right)^{\beta} \quad (18)$$

The partial derivative of equation (16) with respect to s is shown in equation (19):

$$L'(s) = e_1 y_1^* \alpha \Omega - e_2 y_2^* \beta \left(\frac{1}{s(1-s)} - \Omega \right) \quad (19)$$

where $\Omega = \frac{1}{1-\alpha} \left(\frac{1}{s} - \frac{\alpha-\beta}{\delta(s\alpha+(1-s)\beta)} \right)$. Therefore, if the ratio between the emission intensity of the firm producing capital goods and that of the firm producing consumption goods is large enough $\left(\frac{e_1}{e_2} > \frac{y_2^* \beta \left(\frac{1}{s(1-s)} - \Omega \right)}{y_1^* \alpha \Omega} \right)$, the decrease of the saving rate would reduce the CO₂ emissions from the economy. In other words, if the cumulative CO₂ emission intensity of the production process of capital goods is much larger than that of the consumption goods, the saving rate decrease would lead to CO₂ emission reduction.”

We agree with your comment that an uncertainty analysis would enrich the manuscript. We have revised this manuscript by adding a quantitative uncertainty analysis. The range of CO₂ emission changes is estimated with the forecasted potential range of saving rate changes in China. We have added a paragraph with the uncertainty analysis, as shown in lines 371-389:

“Uncertainty in the results of this study mainly comes from the uncertainties of the forecasted saving rate changes of China and the global MRIO data. This study estimates the upper and lower ranges of the CO₂ emission changes based on the forecasted range of potential saving rate changes in China. The range for global industrial CO₂ emission reduction is 292.4–513.3 Mt, while the range for CO₂ emissions increments of Chinese households is 124.3–218.2 Mt. Thus, the range of total CO₂ emissions reduction caused by saving rate changes is 168.1–295.1 Mt (more results in Table S10). Improving the accuracy of the forecasting for saving rates in China would reduce the uncertainties of this study. Meanwhile, the data quality of various global MRIO databases varies due to different compiling methods²⁶. Harmonizing the data quality of various global MRIO databases would reduce the uncertainties of this study. Furthermore, the dynamic consequences of investment for the technological progress of energy conservation fall beyond the scope of this study. Studying the dynamic impacts of technological progress on CO₂ mitigation due to saving rate changes is an interesting future research direction^{27,28}. In addition, due to the limitations of statistical data, the data for China’s inter-regional trade in the MRIO table used in this study are from estimations based on inter-provincial railway transportation data^{13,29}. In the future, estimating inter-regional trade with domestic trade surveys can steadily increase the accuracy of the calculations.”

We also added a paragraph to describe the methods used to evaluate the uncertainties and the sensitivity of the model, as shown in lines 603-615:

“To evaluate the uncertainty of this study, we calculate the range of the CO₂ emission changes with the forecasted range of potential saving rate changes in China. It has been forecasted that the saving rates of China will continuously decrease from 2019 to 2035, thus the upper range of saving rate changes is calculated with the forecasted lowest possible saving rate of China during this period⁹. For the lower range, we assume that the saving rate of China will not be higher than the lowest saving rate in the history after the complete reform of market economy in China (from 1994 to 2019)⁹.

To investigate the sensitivity of this model, we define the sensitivity coefficient s as the ratio between the changing rate of global CO₂ emission reduction and the changing rate of saving rate changes in China. Since the CO₂ emission changes are in a linear correlation with the saving rate changes in the model developed in this study, the sensitivity coefficient of this model equals to 1.”

References:

- 9 Chen, Y., Guo, Y. & Yao, Y. The impact of population aging on high savings rate. *J. Financ. Res.*, 71-84, 2014.
- 13 Mi, Z. et al. Chinese CO₂ emission flows have reversed since the global financial crisis. *Nat. Commun.* 8, 1712, 2017.
- 26 Inomata, S. & Owen, A. Comparative evaluation of MRIO databases. *Econ. Syst. Res.* 26, 239-244, 2014.
- 27 Awaworyi Churchill, S., Inekwe, J., Smyth, R. & Zhang, X. R&D intensity and carbon emissions in the G7: 1870–2014. *Energ. Econ.* 80, 30-37, 2019.
- 28 Jiao, J., Jiang, G. & Yang, R. Impact of R&D technology spillovers on carbon emissions between China’s regions. *Struct. Change Econ. D.* 47, 35-45, 2018.
- 29 Mi, Z. et al. A multi-regional input-output table mapping China's economic outputs and interdependencies in 2012. *Sci. Data* 5, 180155, 2018.

Minor comments:

1. Is there a specific reason to choose the GTAP database? Because there are alternatives such as the Eora database (which also includes the regional MRIO database).

Answer:

Thank you for your comment. We use the GTAP database because of its adequate classification of regions and sectors. GTAP covers 140 regions, which is more than most of the global MRIO databases. Eora covers 190 regions, but its sectoral classification is heterogeneous, which impedes the sectoral-level comparative analyses. Eora has a

harmonized version with a 26-sector classification, but the GTAP has 57 sectors. Given that regional and sectoral aggregations have large impacts on the accuracy of MRIO-based results⁴¹, we choose the GTAP MRIO data in this study. We have revised the Methods section to justify the use of GTAP, as shown in lines 481-492:

“We use the Global Trade Analysis Project (GTAP) MRIO data to construct the EE-MRIO model, because of its adequate classification of regions and sectors. GTAP (including 140 regions) covers more regions than most of the global MRIO databases such as the OECD Inter-Country Input-Output Tables (64 regions), EXIOBASE (49 regions), and World Input-Output Database (43 regions). Although Eora (including 190 regions) covers more regions than GTAP, its sectoral classification is heterogeneous, which cannot support the comparative analyses at the sector level. Eora has an aggregated version with a harmonized 26-sector classification, but the sector resolution of GTAP is higher (57 sectors). In general, given that regional and sectoral aggregations have large impacts on the accuracy of MRIO-based results⁴², we choose the GTAP MRIO data in this study.”

References:

42 Lenzen, M. Aggregation versus disaggregation in input–output analysis of the environment. *Econ. Syst. Res.* 23, 73-89, 2011.

2. The recommendations provided in the Discussion section could have been supported with relevant references.

Answer:

Thank you for your comment. We have revised the Discussion section by providing relevant references which can support the policy implications, as shown in lines 348-353:

“China should take more initiatives regarding the CO₂ mitigation effects of the consumption promotion strategy²¹ and the accelerating process of urbanization^{20,22}. China should further promote the consumption of low-carbon products in the categories of foods, plastic products, electronic equipment, machinery, and electricity, because these products drive massive amounts of CO₂ emissions within global supply chains^{23,24}.”

Lines 360-361:

“China is a large country with great regional disparity^{13,25}. Thus, region-specific policies are important for China¹⁸.”

Lines 382-385:

“Furthermore, the dynamic consequences of investment for the technological progress of energy conservation fall beyond the scope of this study. Studying the dynamic impacts of technological progress on CO₂ mitigation due to saving rate changes is an interesting future research direction^{27,28}.”

References:

- 13 Mi, Z. et al. Chinese CO₂ emission flows have reversed since the global financial crisis. *Nat. Commun.* 8, 1712, 2017.
- 18 Wiedenhofer, D. et al. Unequal household carbon footprints in China. *Nat. Clim. Chang.* 7, 75-80, 2017.
- 20 Feng, K., Hubacek, K., Sun, L. & Liu, Z. Consumption-based CO₂ accounting of China's megacities: The case of Beijing, Tianjin, Shanghai and Chongqing. *Ecol. Indic.* 47, 26-31, 2014.
- 21 Zhang, H., Lahr, M. L. & Bi, J. Challenges of green consumption in China: a household energy use perspective. *Econ. Syst. Res.* 28, 183-201, 2016.
- 22 Wang, Z., Cui, C. & Peng, S. How do urbanization and consumption patterns affect carbon emissions in China? A decomposition analysis. *J. Clean. Prod.* 211, 1201-1208, 2019.
- 23 Liang, S., Qu, S., Zhu, Z., Guan, D. & Xu, M. Income-based greenhouse gas emissions of nations. *Environ. Sci. Technol.* 51, 346-355, 2017.
- 24 Hertwich, E. G. & Peters, G. P. Carbon Footprint of Nations: A Global, Trade-Linked Analysis. *Environ. Sci. Technol.* 43, 6414-6420, 2009.
- 25 Feng, K. et al. Outsourcing CO₂ within China. *P. Natl. Acad. Sci. USA* 110, 11654, 2013.
- 27 Awaworyi Churchill, S., Inekwe, J., Smyth, R. & Zhang, X. R&D intensity and carbon emissions in the G7: 1870–2014. *Energ. Econ.* 80, 30-37, 2019.
- 28 Jiao, J., Jiang, G. & Yang, R. Impact of R&D technology spillovers on carbon emissions between China's regions. *Struct. Change Econ. D.* 47, 35-45, 2018.

3. There are a few typos, which requires a spell-check. For example, the United States of American (USA) on page 8.

Answer:

Thank you for your comment. We have thoroughly checked the spelling of the revised manuscript. We have revised the sentences according to your comments, as shown in lines 137-140:

“For other nations, China’s saving rate increments increased CO₂ emissions of Japan (2.7 Mt), South Africa (1.9 Mt), Australia (1.8 Mt), South Korea (1.7 Mt), India (1.5 Mt), and the United States of America (USA) (1.3 Mt), if other factors remained constant.”

REVIEWERS' COMMENTS:

Reviewer #1 (Remarks to the Author):

The manuscript has substantially improved. Well done to the authors. I would like the authors to elaborate more on their sensitivity analysis. Please see the attached document for more information.

Reviewer's report for

Saving less in China facilitates global CO₂ mitigation

Suggested revision: revision, more in-depth sensitivity analysis

General impression since last review:

The authors have addressed most of my comments and justified why this piece of research should be published in Nature Communications. Well done!

I would still like to ask the authors to elaborate more on their sensitivity analysis.

Justification of opinion:

For some of the results that are presented in this paper the author provided ranges. I assume that these ranges refer to the standard deviations of the calculated values. However, there are still many values, especially small values such as 0.01 Mt that are given and discussed without considering any reliability of the results. During my first review I pointed out that given the relative uncertainty of the individual transaction values in the MRIO framework, a result that is close to 0 may also have the opposite sign or be zero itself. Hence, any conclusion drawn from small results would need to be discussed in the light of a thorough sensitivity analysis.

The authors have justified their sensitivity analysis, but I am not convinced that the reliability data of the underlying MRIO have been considered.

Running Leontief-based footprint analyses requires millions of floating-point operations, especially the inversion of the (I-A) matrix itself. The authors argue that the A matrix is considered as correct, but I disagree that this assumption can hold if other values in the calculation process are viewed in the light of their reliability. I would suggest that the authors undertake an error-propagation-based sensitivity analysis. Further, the underlying GTAP database does not include standard deviation values for its individual transaction values. I would like to ask the authors to describe how they assessed the reliability of the GTAP database. An example for error propagation in input-output calculations can be found in the SI of (Oita et al., 2016).

References

Oita, A., Malik, A., Kanemoto, K., Geschke, A., Nishijima, S., & Lenzen, M. (2016). Substantial nitrogen pollution embedded in international trade. *Nature Geoscience*, 9, 111-115. doi:10.1038/ngeo2635

<http://www.nature.com/ngeo/journal/vaop/ncurrent/abs/ngeo2635.html#supplementary-information>

Reviewer #2 (Remarks to the Author):

Thanks for the revised article. The authors have addressed all of my comments. Hence, from my point of view, the revised article is now appropriate for publication.

Responses to reviewers' comments

Reviewer: 1

The manuscript has substantially improved. Well done to the authors. I would like the authors to elaborate more on their sensitivity analysis. Please see the attached document for more information.

Suggested revision: revision, more in-depth sensitivity analysis

General impression since last review:

The authors have addressed most of my comments and justified why this piece of research should be published in Nature Communications. Well done!

I would still like to ask the authors to elaborate more on their sensitivity analysis.

Justification of opinion:

For some of the results that are presented in this paper the author provided ranges. I assume that these ranges refer to the standard deviations of the calculated values. However, there are still many values, especially small values such as 0.01 Mt that are given and discussed without considering any reliability of the results. During my first review I pointed out that given the relative uncertainty of the individual transaction values in the MRIO framework, a result that is close to 0 may also have the opposite sign or be zero itself. Hence, any conclusion drawn from small results would need to be discussed in the light of a thorough sensitivity analysis.

The authors have justified their sensitivity analysis, but I am not convinced that the reliability data of the underlying MRIO have been considered.

Running Leontief-based footprint analyses requires millions of floating-point operations, especially the inversion of the (I-A) matrix itself. The authors argue that the A matrix is considered as correct, but I disagree that this assumption can hold if other values in the calculation process are viewed in the light of their reliability. I would suggest that the authors undertake an error-propagation-based sensitivity analysis. Further, the underlying GTAP database does not include standard deviation values for its individual transaction values. I would like to ask the authors to describe how they assessed the reliability of the GTAP database. An example for error propagation in input-output calculations can be found in the SI of (Oita et al., 2016).

References

Oita, A., Malik, A., Kanemoto, K., Geschke, A., Nishijima, S., & Lenzen, M. (2016). Substantial nitrogen pollution embedded in international trade. *Nature Geoscience*, 9, 111-115. doi:10.1038/ngeo2635

Answer:

Thank you very much for your comment.

According to your suggestion, we have added an in-depth sensitivity analysis using the matrix-based method considering all the parameters of the model developed in this study⁴⁶. The sensitivity coefficients of this method are equivalent to the coefficients on the variance of the independent parameters in the error-propagation method^{26,47}, which show the partial effects of the changes in parameters on the changes in results. We also define the dimensionless elasticities of the parameters, which indicate the ratios between the changing rate of global CO₂ emission changes and the changing rate of the parameters, to further eliminate the effect caused by the statistical units of the parameters. The results show that the sensitivity of the results in this study to the parameters in this model is low. The parameter with the highest sensitivity is the CO₂ emission intensity of the Metallurgy sector in Hebei (elasticity 0.14).

We have revised the manuscript by adding the contents regarding sensitivity analysis, as shown in lines 355-368:

“We evaluate the sensitivity of the results to all the parameters by calculating their elasticities, which are the ratios between the changing rate of global CO₂ emission changes and the changing rate of the parameters. Results show that the elasticities of most parameters are small (Supplementary Figure 6), which indicates a low sensitivity for the results. The parameter with the highest sensitivity is the CO₂ emission intensity of the Metallurgy sector in Hebei (elasticity 0.14), which means that the change in CO₂ emissions from global production systems due to the change in saving rates would change by 1.4% if the CO₂ emission intensity of the Metallurgy sector in Hebei changes by 10%. Other parameters with relatively high sensitivity include the disposable incomes and saving rate changes of Shandong and Jiangsu. The CO₂ emissions of Chinese households are in a linear correlation with the saving rate changes, in which the saving rate change in Inner Mongolia has the largest sensitivity (elasticity 0.09). Full results of sensitivity analysis for all the parameters are given in Supplementary Tables 11-20.”

We have also added the method for the sensitivity analysis in the Methods section, as shown in lines 591-636:

“We use the matrix-based method considering all the parameters of the model developed in this study to investigate the sensitivity of this model⁴⁶. We first calculate the sensitivity coefficient as the change of the CO₂ emissions caused by a marginal change in each of the parameters. The sensitivity coefficients of this method are equivalent to the coefficients on the variance of the independent parameters in the error-propagation method^{26,47}, which show the partial effects of the changes in parameters on the changes in results. For global industrial CO₂ emissions, the sensitivity coefficients for the CO₂ emission intensity of nation sector j , each element T_{ij} in the intermediate transaction matrix T of the MRIO table, the disposable income (W_m) of Chinese region m , the consumption structure ($s_{c,m}$) of Chinese region m , the capital formation structure ($s_{i,m}$) of Chinese region m , and the

saving rate change α , are calculated with equations (40) to (45), respectively. The notation Δe^p represents the changes of global industrial CO₂ emissions due to the changes in saving rates of China; $s_{c,mj}$ and $s_{i,mj}$ represent the proportions of nation sector j in the final demand and capital formation of region m , respectively; and x_j indicates the total output of nation sector j .

$$\frac{\partial \Delta e^p}{\partial q_j} = [L(f^* - f)]_j \quad (40)$$

$$\frac{\partial \Delta e^p}{\partial T_{ij}} = \frac{(qL)_i [L(f^* - f)]_j}{x_j} \quad (41)$$

$$\frac{\partial \Delta e^p}{\partial W_m} = qL\alpha(s_{i,m} - s_{c,m}) \quad (42)$$

$$\frac{\partial \Delta e^p}{\partial s_{i,mj}} = (qLW_m\alpha)_j \quad (43)$$

$$\frac{\partial \Delta e^p}{\partial s_{c,mj}} = -(qLW_m\alpha)_j \quad (44)$$

$$\frac{\partial \Delta e^p}{\partial \alpha} = \sum_m [qLW_m(s_{i,m} - s_{c,m})] \quad (45)$$

For CO₂ emissions from Chinese households, the sensitivity coefficient for the saving rate change α is calculated by equation (46):

$$\frac{\partial \Delta e_m^h}{\partial \alpha} = -\left(\frac{e_m^h}{1-r_m}\right) \quad (46)$$

where Δe_m^h represents the change of CO₂ emissions from households in Chinese region m due to the changes in saving rates of China, and r_m indicates the current saving rate of Chinese region m .

To further eliminate the effect caused by the statistical units of the parameters, we define the dimensionless elasticities of the parameters, which indicate the ratios between the changing rate of global CO₂ emission changes and the changing rate of the parameters. The elasticities are calculated by equations (47) to (53):

$$EL_{q_j}^p = \frac{\partial \Delta e^p}{\partial q_j} \times \frac{q_j}{\Delta e^p} \quad (47)$$

$$EL_{T_{ij}}^p = \frac{\partial \Delta e^p}{\partial T_{ij}} \times \frac{T_{ij}}{\Delta e^p} \quad (48)$$

$$EL_{W_m}^p = \frac{\partial \Delta e^p}{\partial W_m} \times \frac{W_m}{\Delta e^p} \quad (49)$$

$$EL_{s_c,mj}^p = \frac{\partial \Delta e^p}{\partial s_{c,mj}} \times \frac{s_{c,mj}}{\Delta e^p} \quad (50)$$

$$EL_{s_i,mj}^p = \frac{\partial \Delta e^p}{\partial s_{i,mj}} \times \frac{s_{i,mj}}{\Delta e^p} \quad (51)$$

$$EL_{\alpha}^p = \frac{\partial \Delta e^p}{\partial \alpha} \times \frac{\alpha}{\Delta e^p} \quad (52)$$

$$EL_{\alpha}^h = \frac{\partial \Delta e^h}{\partial \alpha} \times \frac{\alpha}{\Delta e^h} \quad (53)$$

where $EL_{q_j}^p$, $EL_{T_{ij}}^p$, $EL_{W_m}^p$, $EL_{s_c,mj}^p$, $EL_{s_i,mj}^p$, and EL_{α}^p represent the elasticities for global industrial CO₂ emissions to the CO₂ emission intensity, the elements of intermediate transaction matrix, the disposal incomes of Chinese regions, the final consumption structure of Chinese regions, the capital formation structure of Chinese regions, and the saving rate changes, respectively. The notation EL_{α}^h denotes the elasticities for the CO₂ emissions of Chinese households to the saving rate changes.”

Meanwhile, we agree with the reviewer’s comment that using standard deviations would help us to evaluate the reliability and uncertainty of the results given by a certain database. Unfortunately, due to the lack of uncertainty information on the raw data from the statistical offices, the standard deviations of the MRIO databases were either unavailable or provided by assumptions and choices²⁷. Given that the GTAP database does not provide standard deviations, in the lack of uncertainty information on raw statistical data, to avoid the assessment of uncertainty by using data with uncertainty, we prefer to do not conduct a quantitative uncertainty analysis by using standard deviations. Despite this, according to the reviewer’s suggestion, we have revised the manuscript regarding this issue.

We have elaborated the uncertainty analysis, as shown in lines 343-354:

“Meanwhile, uncertainties of the global MRIO data would also cause uncertainties in the results. An uncertainty analysis with standard deviations can be used to evaluate the reliability and uncertainty of the results given by a certain database²⁶. However, the standard deviations are not given in the GTAP database, which makes the condition insufficient for quantitative uncertainty analysis. Besides, unfortunately, due to the lack of uncertainty information on the raw data from the statistical offices, the standard deviations of the MRIO databases are either unavailable or provided by assumptions and choices²⁷. If the statistical offices could publish more information on the data variations of their original samples and the degrees of uncertainty in their data processing stages, future work with MRIO models could provide a more precise uncertainty analysis.”

Meanwhile, we have further justified the reliability of the GTAP database, as shown in lines 464-472:

“We use the Global Trade Analysis Project (GTAP) MRIO data to construct the EE-MRIO model because the GTAP database has relatively high resolutions of nations and comparable sectors. GTAP database has a quality control process to prioritize the usage of data sources with higher degrees of reliability⁴³. The comparison studies for MRIO databases show that GTAP produces similar results with Eora⁴⁴ and WIOD⁴⁵ for the majority of regions. Meanwhile, GTAP also has a policy of placing a premium on the continuity of data suppliers⁴³. This property makes it suitable for the structural decomposition analysis using two GTAP MRIO tables.”

References:

26 Oita, A. et al. Substantial nitrogen pollution embedded in international trade. *Nat. Geosci.* 9, 111-115 (2016).

27 Lenzen, M., Moran, D., Kanemoto, K. & Geschke, A. Building EORA: A global multi-region input–output database at high country and sector resolution. *Econ. Syst. Res.* 25, 20-49 (2013).

43 Aguiar, A., Chepeliev, M., Corong, E. L., McDougall, R. & van der Mensbrugge, D. The GTAP Data Base: Version 10. *J. Glob. Econ. Anal.* 4, 27 (2019).

44 Wieland, H., Giljum, S., Bruckner, M., Owen, A. & Wood, R. Structural production layer decomposition: a new method to measure differences between MRIO databases for footprint assessments. *Econ. Syst. Res.* 30, 61-84 (2018).

45 Owen, A., Steen-Olsen, K., Barrett, J., Wiedmann, T. & Lenzen, M. A structural decomposition approach to comparing MRIO databases. *Econ. Syst. Res.* 26, 262-283 (2014).

46 Heijungs, R. Sensitivity coefficients for matrix-based LCA. *Int. J. Life. Cycle. Ass.* 15, 511-520 (2010).

47 Heijungs, R. & Lenzen, M. Error propagation methods for LCA—a comparison. *Int. J. Life. Cycle. Ass.* 19, 1445-1461 (2014).

Reviewer #2 (Remarks to the Author):

Thanks for the revised article. The authors have addressed all of my comments. Hence, from my point of view, the revised article is now appropriate for publication.

Answer:

Thank you very much!